# Impact of mountain-wave-induced temperature fluctuations on the occurrence of polar stratospheric ice clouds: A statistical analysis based on MIPAS observations and ERA5 data

Ling Zou[1,3], Reinhold Spang[2,3], Sabine Griessbach[1,3], Lars Hoffmann[1,3], Farahnaz Khosrawi[1,3], Rolf Müller[2,3], and Ines Tritscher[2,3]

[1]Jülich Supercomputing Centre (JSC), Forschungszentrum Jülich, Jülich, Germany
[2]Institute of Energy and Climate Research (IEK-7), Forschungszentrum Jülich, Jülich, Germany
[3]Center for Advanced Simulation and Analytics (CASA), Forschungszentrum Jülich, Jülich, Germany

**Correspondence:** Ling Zou (l.zou@fz-juelich.de), Lars Hoffmann (l.hoffmann@fz-juelich.de)

**Abstract.** Temperature fluctuations induced by mountain waves can play a crucial role in the formation of polar stratospheric clouds (PSCs). In particular, the cold phase of the waves can lower local temperatures sufficiently to trigger PSC formation even when large-scale background temperatures are too high. To provide new quantitative constraints on the relevance of this effect, this study analyzes a decade ($2002-2012$) of ice PSC detections obtained from Michelson Interferometer for Passive Atmospheric Sounding (MIPAS/Envisat) measurements and ERA5 reanalysis data in the polar winter lower stratosphere. In the MIPAS observations, we find that approximately $52\%$ of the Arctic and $26\%$ of the Antarctic ice PSCs are detected at temperatures above the local $T_{ice}$. Ice PSCs above $T_{ice}$ are concentrated around mountainous regions and their downwind directions. A backward trajectory analysis is performed to investigate the temperature history of each ice PSC observation. The cumulative fraction of ice PSCs above $T_{ice}$ increases as the trajectory gets closer to the observation point. The most significant change of the fraction of ice PSCs above $T_{ice}$ occurs within the $6\,\mathrm{h}$ preceding the observations. At the observation point, the mean fractions of ice PSCs above $T_{ice}$, taking into account temperature fluctuations along the backward trajectory, are $33\%$ in the Arctic and $9\%$ in the Antarctic. The results provide a quantitative assessment of the occurrence of ice PSCs above $T_{ice}$ in connection with orographic waves. Additionally, the observational statistics presented can be utilized for comparison with chemistry-climate simulations.

## 1 Introduction

Polar stratospheric clouds (PSCs) form in winter and early spring in the polar stratosphere. Supercooled ternary solution (STS) droplets, nitric acid trihydrate (NAT), and ice particles are the three main types of PSCs. The surface of PSC particles enables heterogeneous chemical reactions that convert reservoir species into chlorine radicals, subsequently leading to ozone depletion in the polar winter stratosphere (Solomon et al., 1986; Solomon, 1999). Denitrification through the sedimentation of large NAT PSC particles also contributes to prolonged ozone depletion, as it reduces the concentration of $NO_2$ while increasing $Cl_2$. The formation processes and conditions are summarized in Lambert et al. (2012) and Tritscher et al. (2021), highlighting the pivotal role of temperature in these processes. In the polar regions, synoptic low temperatures occur in winter and spring, causing temperatures to drop well below the formation thresholds and resulting in substantial formation of ice PSCs (Campbell and Sassen, 2008).

Atmospheric gravity waves are oscillations in the Earth's atmosphere caused by buoyancy and gravity acting on air parcels displaced from their equilibrium positions. They can significantly affect local pressure, temperature, winds, and other meteorological variables. Gravity waves play a critical role in the transfer of energy and momentum between different layers of the atmosphere, i.e., they can significantly affect the temperature structure and general circulation of the atmosphere (Fritts and Alexander, 2003; Alexander et al., 2010). Gravity waves typically originate from sources such as mountain ranges, where air flow is disturbed and lifted; thunderstorms, where convective activity is generated; and frontal systems, where air masses are disturbed from geostrophic equilibrium. Understanding gravity waves is essential for accurate weather forecasting and broader atmospheric dynamics. Properly representing the effects of gravity waves in general circulation and chemical transport models is challenging because coarse-grid global models typically lack the spatial resolution needed to resolve the small-scale perturbations of gravity waves. In this study, we are particularly interested in atmospheric gravity waves in the polar winter lower stratosphere, because temperature perturbations due to gravity waves can trigger the development of PSCs in the cold phase of these waves, even in cases where background or synoptic-scale temperatures are higher than their formation thresholds (Carslaw et al., 1998b; Rivière et al., 2000; Dörnbrack et al., 2020; Orr et al., 2020). Missing these waves in coarse-resolution global chemistry-climate simulations or reanalysis-driven chemistry-transport simulations will affect the PSC representation.

PSCs triggered by small-scale temperature fluctuations have been observed predominantly in various mountain regions, including Greenland and Scandinavia in the Arctic, as well as the Antarctic Peninsula and the Transantarctic Mountains in the Antarctic. For example, over the Antarctic Peninsula, the Cloud-Aerosol Lidar Infrared Pathfinder Satellite Observations (CALIPSO) detected 63 % of ice PSC volumes during gravity wave events during the 2006 – 2010 Antarctic winters (Noel and Pitts, 2012). The mountain waves were also verified by Höpfner et al. (2006) and Eckermann et al. (2009) as a trigger for the appearance of the Antarctic NAT belt, originating from ice particles over the Antarctic Peninsula in June 2003, as evidenced by measurements from the Michelson Interferometer for Passive Atmospheric Sounding (MIPAS). Analyses based on CALIPSO data revealed that 75 % of ice PSCs and 50 % of NAT mixtures during the Antarctic (2007–2010) and Arctic (2006/2007 to 2009/2010) winter seasons were closely linked to mountain wave activity (Alexander et al., 2013). Due to the importance of wave-generated ice PSCs, Pitts et al. (2013) introduced an additional class in their PSC characterisation scheme specifically for

wave ice PSCs. The pivotal role of gravity waves, which result in small-scale temperature fluctuations triggering the formation of PSCs, has been discussed in WMO (2018). Additionally, WMO (2022) has addressed that the occurrence of PSCs over the Antarctic Peninsula is linked to frequent mountain wave activities.

By tracking the trajectories of air parcels, Lagrangian models provide insights into the nucleation, growth, sedimentation, and evaporation of PSC particles, offering a detailed view of the formation and evolution of PSCs (Santee et al., 2002; Lambert et al., 2012; Hoffmann et al., 2017b; Tritscher et al., 2019). By combining satellite observations with trajectory modeling, Wegner et al. (2013) have been able to assess phenomena like vortex-wide chlorine activation by mesoscale PSC events, showcasing the utility of this approach in studying complex atmospheric processes. More recently, the Hybrid Single-Particle Lagrangian Integrated Trajectory dispersion model (HYSPLIT) has been employed to perform trajectory calculations, aiding in understanding the widespread presence of PSCs during certain Arctic winters (Voigt et al., 2018). The Chemical Lagrangian Model of the Stratosphere (CLaMS) is also capable of simulating the intricate processes involved in PSC development along individual trajectories (Tritscher et al., 2019). Here, we are particularly interested in using Lagrangian trajectory analyses for assessing the temperature history along backward trajectories from ice PSC observations. Based on multiple Arctic vortex trajectories, Carslaw et al. (1999) have unveiled the significance of temperature perturbations induced by mountain waves in generating solid PSC particles in the east of Greenland, the Norwegian mountains, and the Urals during December 1994 and January 1995. However, the small-scale temperature fluctuations related to mountain waves are often underestimated or not fully resolved in global reanalyses or coarse-resolution chemistry-climate models (Orr et al., 2015; Hoffmann et al., 2017b; Orr et al., 2020; Weimer et al., 2021).

In this study, our primary focus is to investigate and provide new quantitative estimates of the occurrence of ice PSCs as observed by the Envisat MIPAS satellite experiment (Spang et al., 2004, 2018; Höpfner et al., 2018) and how they relate to atmospheric temperatures above the ice existence threshold ($T_{\text{ice}}$) as derived from the European Centre for Medium-Range Weather Forecasts (ECMWF) ERA5 reanalysis (Hersbach et al., 2020). Assessing the fraction of MIPAS ice PSC detections at synoptic scale temperatures well above the ERA5 derived frost point temperature helps to characterize the crucial role of unresolved or only poorly resolved subgrid-scale temperature fluctuations due to gravity waves in the ERA5 reanalysis. This study provides quantitative information on the effects of unresolved temperature fluctuations in the ERA5 reanalysis, which is of particular interest for evaluating chemistry-transport or climate model simulations of polar stratospheric ozone.

Here, we also propose a new method to better locate the MIPAS ice PSC observations. Previous studies using MIPAS data usually assumed that PSCs are homogeneously stratified and located at the tangent point of the instrument's line of sight. Considering the broad vertical field of view of MIPAS ($3-4\,\text{km}$) and the potential variability of PSC occurrence and composition within the field of view and along the line of sight, we propose to consider the location of the minimum of the temperature difference of $T - T_{\text{ice}}$ as the most likely location of the PSC. Here, ice PSCs are analyzed at temperature-based sampling points during the Arctic (November–February) and Antarctic (June–September) winters, covering the full-time period of MIPAS measurements from 2002 to 2012. Additionally, leveraging the capabilities of the Massive-Parallel Trajectory Calculations (MPTRAC) model (Hoffmann et al., 2016, 2022) developed at the Jülich Supercomputing Centre, we

explore the temperature variations along the backward trajectories driven by ERA5 data as an indicator of gravity wave activity to comprehend their impact on ice PSC occurrence.

The study begins with a comprehensive overview of the data and methodology, detailing the retrieval of ice PSCs from MIPAS, introducing the MPTRAC model, and describing the detection of temperature fluctuations (Sect. 2). Spatio-temporal features of ice PSCs are analyzed in Sect. 3.1, followed by an examination of ice PSC observations above $T_{ice}$ in Sect. 3.2. The Lagrangian history and temperature fluctuations in connection with ice PSC observations above $T_{ice}$ are explored in Sect. 3.3 and 3.4, respectively. The occurrence of ice PSC above $T_{ice}$ in connection with mountain waves is investigated in Sect. 3.5. Uncertainties inherent in the study are discussed in Sect. 4, while the main conclusions drawn from the analysis are summarized in Sect. 5.

## 2 Data and method

### 2.1 MIPAS observations of PSCs

The Michelson Interferometer for Passive Atmospheric Sounding (MIPAS) on board the Envisat satellite measured limb infrared spectra in the $4 - 15\,\mu$m wavelength range with high-resolution from the mid-troposphere to the mesosphere (Fischer et al., 2008). The Envisat satellite, which was in a sun-synchronous low Earth orbit (98.4° inclination), was able to capture measurements with coverage up to both poles, attributable to an extra poleward tilt of the primary mirror. From 2002 to 2004, MIPAS collected samples with a resolution of 3 km (vertical) $\times$ 30 km (horizontal) at the tangent point. Later, between January 2005 and April 2012 the vertical sampling below 21 km was optimized to 1.5 km in the nominal measurement mode. MIPAS ceased operation on 8 April 2012 due to the sudden loss of contact with Envisat.

Spang et al. (2001) introduced a simple and reliable method for detecting clouds in infrared limb sounder measurements by comparing the mean radiances of two different spectral wavelength regions. Each region responds differently to clouds in the field of view. The first region, $788 - 796\,$cm$^{-1}$, is primarily influenced by $CO_2$ emissions and shows little change in the presence of optically thin clouds. In contrast, the second region, $832 - 834\,$cm$^{-1}$, is located in an atmospheric window region and mainly influenced by aerosol and cloud emissions. The ratio of radiances, called the cloud index (CI), is high for cloud-free conditions (CI $> 4$), close to one for optically thick conditions, and falls in between for the transition from optically thin to thick clouds.

The CI is known to be sensitive to PSCs (Spang et al., 2004). A fast prototype processor for retrieving cloud parameters from MIPAS (MIPclouds) is described in Spang et al. (2012), where PSC detections and their cloud top heights are obtained by a step-like data processing approach of up to five detection methods, including the CI. More than 600,000 modeled MIPAS-like spectra are included to represent PSC composition (Spang et al., 2012). A Bayesian probabilistic scheme identifies the different types of PSCs in MIPAS measurements based on the combination of CI, NAT-index (NI), and brightness temperature differences (Spang et al., 2016). Eight classes (-1: unclassified (non-cloudy), 0: unknown, 1: ice, 2: NAT, 3: STS, 4: ICE_NAT, 5: STS_NAT, and 6: ICE_STS) are defined based on the normalized product probability for each spectrum (Spang et al., 2016, 2018). Class 1-3 are pure type classes, while class 4-6 are mixed type classes.

In this study, ice PSCs were extracted from the MIPAS/Envisat Observations of Polar Stratospheric Clouds dataset (Spang, 2020). Typically, the tangent point, which is the point closest to the Earth's surface along the line of sight, serves as the reference point for MIPAS observations of PSC locations (Fischer et al., 2008). However, this is not optimal for cloud observations in the limb, where the precise location of the cloud along the line of sight remains uncertain. Depending on the atmospheric conditions, the cloud's location could deviate by several hundred kilometers horizontally, either in front or behind the tangent point. Therefore, instead of using the tangent point of the sample, we employ the point where the temperature difference ($\Delta T_{\mathrm{ice\_min}}$) between the frost point temperature ($T_{\mathrm{ice}}$) and the environmental temperature along the line of sight ($T_{\mathrm{LoS}}$) is minimal:

$$\Delta T_{\mathrm{ice\_min}} = \min(T_{\mathrm{LoS}} - T_{\mathrm{ice}}) \tag{1}$$

Equation 1 was evaluated up to a maximum altitude of 30 km. Assigning the ice PSC observations to $\Delta T_{\mathrm{ice\_min}}$, instead of the tangent point, places them at the most favorable and probable condition along the line of sight. Here, $T_{\mathrm{ice}}$ and the temperature along the line of sight were derived from meteorological reanalysis data (Sect. 2.3).

In addition, only data with a CI > 1.2 (not completely optically thick) and CI < 4 (detection threshold) were used in this study. All detected ice PSCs met the conditions for potential vorticity $\geq 4$ PVU, heights between 14 and 30 km, and only at an altitude of 6.1 km below the cloud top. All of the above criteria helped to ensure that measurements are located in the stratosphere and that potentially optically thick cases were excluded. Northern hemispheric (NH) PSCs were based on data for the Arctic winter (December – February), and Southern hemispheric (SH) PSCs were based on data for the Antarctic winter (June – September) at latitudes higher than 50°N/S from December 2002 to February 2012. Data from 2002, 2004 in the Antarctic and 2003 in the Arctic were excluded due to many missing observations.

## 2.2 The MPTRAC model

Lagrangian particle dispersion models can precisely represent atmospheric transport processes by computing air parcel trajectories. The MPTRAC model (Hoffmann et al., 2016, 2022) was developed to study large-scale atmospheric transport in the free troposphere and stratosphere. The MPTRAC model includes a variety of modules and tools, e.g., modules of advection, turbulent diffusion, subgrid-scale wind fluctuations, convection, sedimentation, wet and dry deposition, hydroxyl chemistry, exponential decay, and boundary conditions. In the advection module, air parcel trajectories are calculated based on given wind fields from meteorological data sets. Following the Flexible Particle dispersion (FLEXPART) model (Stohl et al., 2005), turbulent diffusion and subgrid-scale wind fluctuations are simulated by MPTRAC by adding stochastic perturbations to the trajectories. The MPTRAC model was applied to calculate trajectories for MIPAS PSC observations (Hoffmann et al., 2017b) and the trajectories calculated from the MPTRAC model were evaluated by superpressure balloons for the polar lower stratosphere (Hoffmann et al., 2017a). In this study, the advection module of MPTRAC is applied to calculate backward trajectories based on PSC observations. The tool for meteorological data sampling is used to obtain corresponding meteorological data for the MIPAS PSC observations, including the temperature, humidity, and frost point along the trajectories (Hoffmann et al., 2022).

## 2.3 Meteorological reanalysis

The fifth-generation reanalysis (ERA5 Hersbach et al., 2020) from ECMWF provides hourly meteorological data with a horizontal resolution of about 31 km on 137 hybrid sigma/pressure levels vertically from the surface to 0.01 hPa. It was used for identifying the location of ice PSCs ($\Delta T_{\text{ice\_min}}$) along the line of sight of MIPAS and calculating $T_{\text{ice}}$ in this study. The trajectory calculations with MPTRAC were also based on ERA5 reanalysis. The MPTRAC trajectory calculations were assessed by using different meteorological reanalyses (Hoffmann et al., 2017b; Rößler et al., 2018; Hoffmann et al., 2019). It was found that Lagrangian transport simulations are significantly improved by the ERA5 data compared to ERA-Interim data (Hoffmann et al., 2019). For instance, it was found that there is better conservation of potential temperature along the ERA5 trajectories than the ERA-Interim trajectories in the lower stratosphere.

## 2.4 Ice PSCs and detection of temperature fluctuations

To assess the impact of temperature fluctuations caused by gravity waves on PSCs we used MIPAS measurements and ERA5 data. First, we identified ice PSC observations where the temperature at the location of the MIPAS observation is above the frost point temperature ($T_{\text{ice}}$). $T_{\text{ice}}$ was calculated from pressure and $H_2O$ derived from ERA5 reanalyses using the equation proposed by Marti and Mauersberger (1993), which is derived from direct measurements of the vapor pressure down to temperatures of 170 K. The ice frost point indicates the temperature threshold below which ice particles can exist.

The variance of the temperature cooling rate over 6 hours was used to identify temperature fluctuations along the kinematic backward trajectory. We empirically identified potentially significant temperature fluctuations using a variance of the temperature cooling rate greater than $0.9\,\text{K}^2\,\text{h}^{-2}$ and a temperature within 10 K above $T_{\text{ice}}$ as the selection criteria. It is important to note that the amplitudes of temperature fluctuations are often underestimated in the ERA5 reanalysis. Therefore, small thresholds are needed to detect potentially relevant wave events.

## 3 Results

### 3.1 Ice PSC observations

The horizontal and vertical distribution of the average occurrence frequency of ice PSCs at $\Delta T_{\text{ice\_min}}$ derived from MIPAS measurements for the period from $2002-2012$ is presented in Fig. 1. Our results of the ice PSC distributions are in general agreement with the respective occurrence frequencies shown in Tritscher et al. (2021). In the Antarctic, ice PSCs are observed at all latitudes south of 65°S, but favor the 90°W to 90°E longitude range with the highest occurrence frequency of up to 16 %. Over the course of a year, ice PSCs are predominantly observed during mid-winter, from late June to September, and are mostly detected in the altitude range of 22 km to 28 km. In the Arctic, ice PSCs are mainly observed within the longitude range of 60°W to 120°E with the highest occurrence frequency of about 2 %, which is a favored region for the locations of the Arctic polar vortex and mountain wave activities (Alexander et al., 2009; Zhang et al., 2016). Note that while the Antarctic polar vortex is centred over the pole, the Arctic polar vortex is displaced from the pole due to frequent disturbances from

180 sudden stratospheric warmings (SSWs) caused by planetary wave activity (Waugh and Randel, 1999; Baldwin et al., 2021). The highest occurrence frequencies of ice PSCs in the Arctic are found in February at an altitude range of 22 km to 26 km (Fig. 1d).

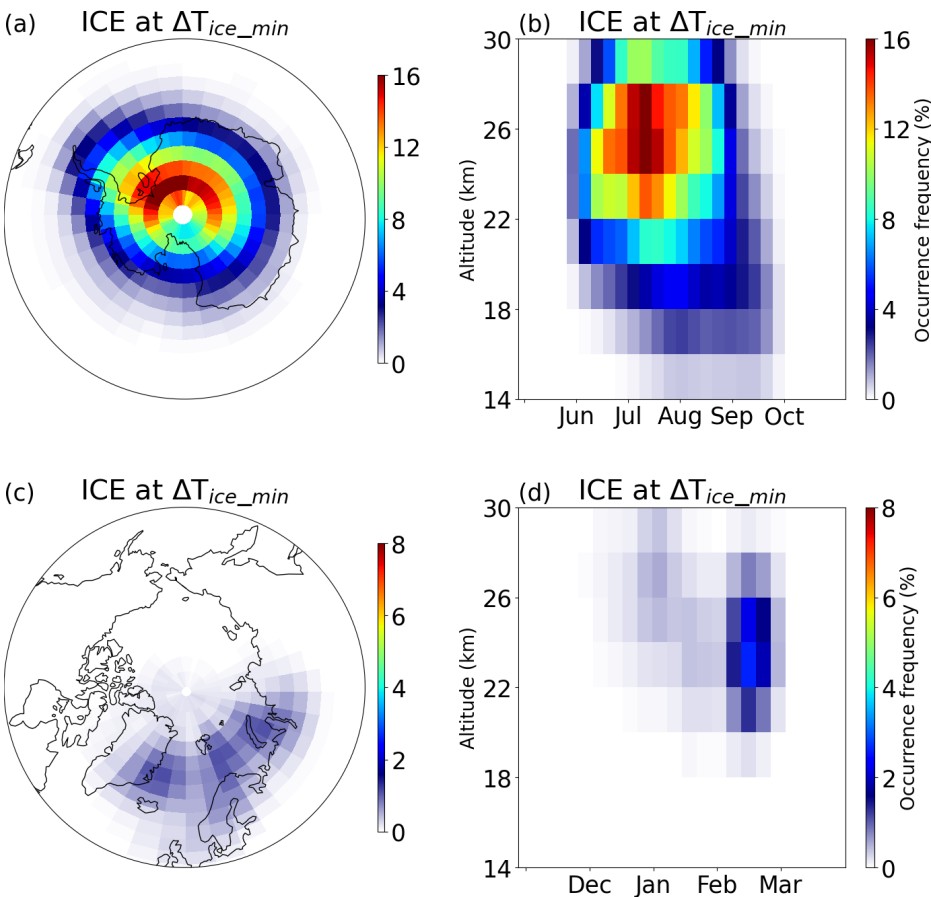

**Figure 1.** Occurrence frequency of ice PSCs (detected PSCs with respect to all measurements) at the point where the temperature difference between $T_{ice}$ and $T$ along the line of sight ($\Delta T_{ice\_min}$) is at its minimum. The ice PSCs were derived from MIPAS observations for the period from 2002 to 2012. The data is gridded on $3° \times 5°$ boxes and the time series are averaged over 7 days and 2 km height bins.

Figure 2 shows the averaged (2002 – 2012) occurrence frequency of ice PSCs as a function of the temperature difference to $T_{ice}$. There are approximately 52 % and 26 % of ice PSCs in the Arctic and Antarctic, respectively, detected at temperatures 185 above the local frost point temperature ($T_{ice}$). Since ice particles nucleate homogeneously at temperatures 3 – 4 K below $T_{ice}$ (Koop et al., 2000), and heterogeneously on solid hydrates at temperatures 0.1 – 1.3 K below $T_{ice}$ (Carslaw et al., 1998a; Koop et al., 1998; Fortin et al., 2003; Engel et al., 2013; Voigt et al., 2018), we included the two temperature thresholds, $T_{ice} - 3$ K for homogeneous nucleation and $T_{ice} - 1.5$ K for heterogeneous nucleation, in addition to the temperature threshold of $T_{ice}$. Regarding nucleation temperatures, more than 90 % of the ice PCSs were observed at a temperature above $T_{ice} - 1.5$ K in both

hemispheres and even 100 % of the ice PSCs were observed at a temperature above $T_{\text{ice}} - 3$ K. This implies that the majority of ice PSCs were above their nucleation temperature, based on the temperature and water vapor data from the ERA5 reanalysis.

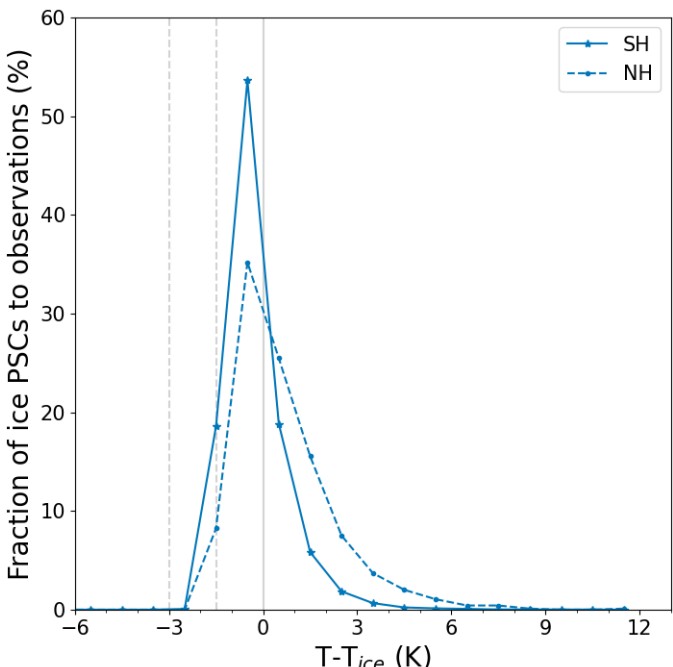

**Figure 2.** Occurrence frequency of ice PSCs as a function of the temperature difference $(T - T_{\text{ice}})$ with a bin size of 1 K. The blue solid line represents data from the Antarctic, while the blue dashed line represents data from the Arctic. The gray solid line indicates the equilibrium temperature, under which ice PSCs can exist. The gray dashed lines indicate the homogeneous and heterogeneous nucleation temperatures, $T - T_{\text{ice}} = -3$ K and $T - T_{\text{ice}} = -1.5$ K, respectively.

## 3.2 Characteristics of ice PSC observations above $T_{ice}$

In the Antarctic, ice PSCs above $T_{ice}$ are predominantly concentrated around the Antarctic Peninsula and its downwind direction with a peak value of about 5 %, notably towards the Weddell Sea (see Fig. 3a). In the Arctic, the spatial distribution pattern of ice PSCs above $T_{ice}$ is similar to that of all the observed ice PSCs, which are distributed across areas such as East Greenland and Scandinavia with peak values of about 1 %, along with their respective downwind directions (see Fig. 3c). The vertical and temporal evolution of ice PSCs above $T_{ice}$ shows that they are mainly concentrated in deep winter, from July to the middle of August in the Antarctic. Ice PSCs above $T_{ice}$ (Fig. 3b) show a decrease in height as late winter approaches, particularly in August and September. A similar pattern is observed in the Arctic, where the polar vortex warms from above, leading to a downward shift of the coldest temperatures later in the winter due to diabatic descent (Rosenfield et al., 1994). Low fractions (about 1 %) of ice PSCs are present in the Arctic, which are similar to CALIOP results (Alexander et al., 2013).

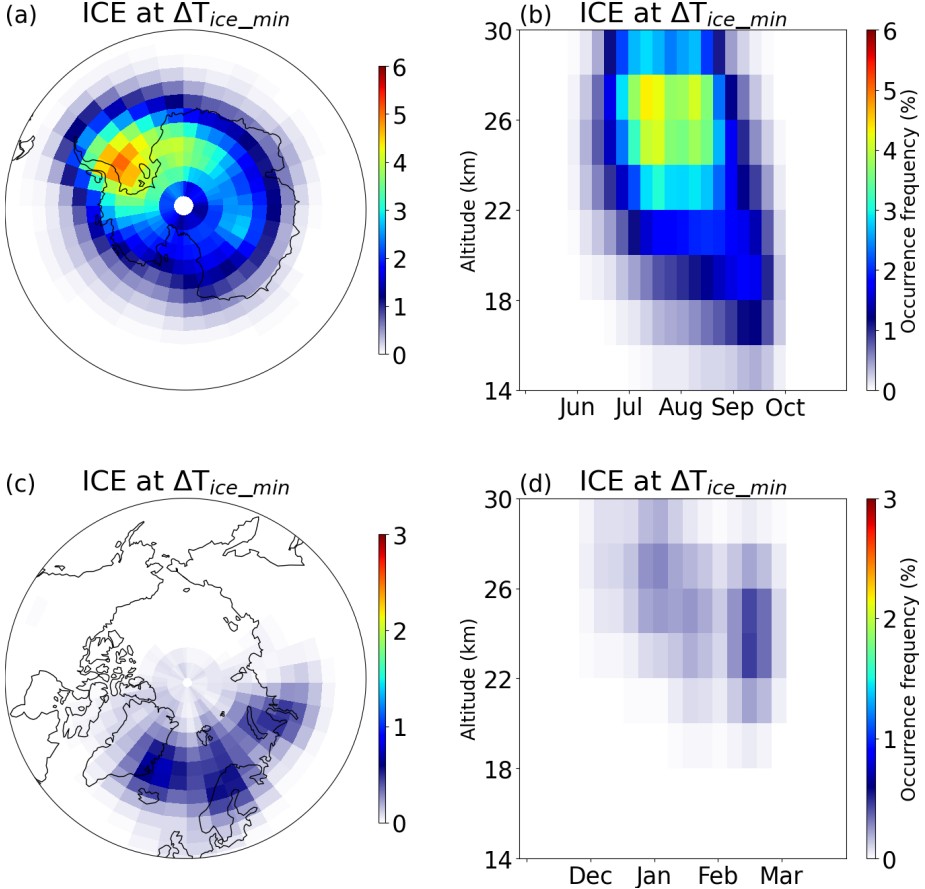

**Figure 3.** Same as Fig. 1, but for the occurrence frequency of ice PSCs above $T_{ice}$ at $\Delta T_{ice\_min}$ only.

Figure 4 shows the fractions of ice PSCs above $T_{ice}$ with respect to all detected PSCs as a function of winter months and years. In the Antarctic (Fig. 4a,b), the fraction of ice PSCs above $T_{ice}$ increases from 15 % in June to 60 % in September, remaining relatively stable across the years. In the Arctic (Fig. 4c,d), the fraction of ice PSCs above $T_{ice}$ is 51 % in January and February and 70 % in December due to less ice PSC observations in this month. However, their fraction shows substantial variability from year to year. Notably, there is a lower occurrence of ice PSCs above $T_{ice}$ in the years 2005 and 2011. The Arctic winters of 2004/2005 and 2010/2011 were both distinguished by exceptionally low temperatures and substantial ozone depletion (e.g. Feng et al., 2007; Manney et al., 2011). Thus, ice PSCs observed in these two winters rather occurred at temperatures below $T_{ice}$ than above $T_{ice}$. The peak value observed in 2008 could be linked to the eruption of the Kasatochi volcano (Waythomas et al., 2010). The larger temperature variability and less stable polar vortex in the Arctic (Newman et al., 2001) compared to the Antarctic cause larger variability in ice PSC occurrence in the Arctic than in the Antarctic. Considering the application of nucleation temperature thresholds ($T_{ice} - 3\,\mathrm{K}$ and $T_{ice} - 1.5\,\mathrm{K}$), the fraction of ice PSCs above $T_{ice} - 3\,\mathrm{K}$ and $T_{ice} - 1.5\,\mathrm{K}$ increases significantly. This fraction exceeds 95 % in both hemispheres.

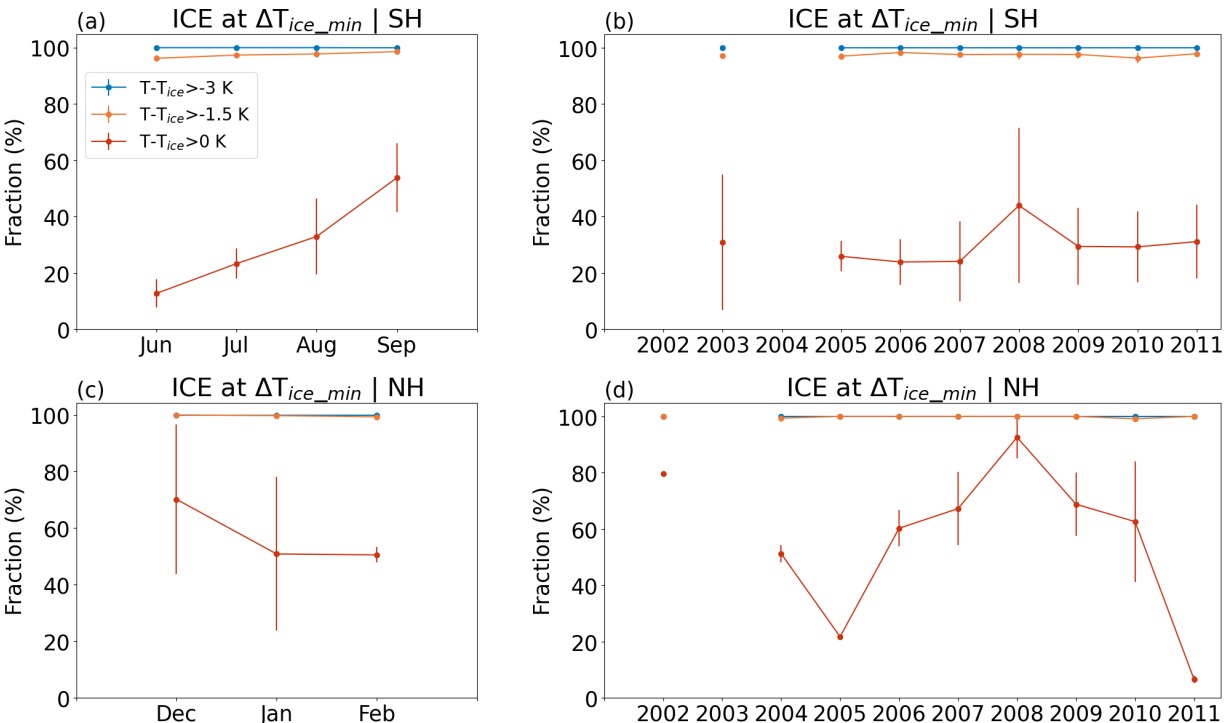

**Figure 4.** Fraction of ice PSCs above $T_{ice}$ to all detected PSCs across different months and years for the Antarctic (a,b) and the Arctic (c, d). Red lines represent the fractions with $T - T_{ice} > 0\,\mathrm{K}$, while orange and blue lines represent the fractions with $T - T_{ice} > -1.5\,\mathrm{K}$ and $T - T_{ice} > -3\,\mathrm{K}$, respectively. The data for the Antarctic in 2002 and 2004, as well as for the Arctic in 2003, are missing due to missing MIPAS observations.

### 3.3 Temperature history of ice PSC observations above $T_{\text{ice}}$

To gain deeper insights into the history of the ice PSC observations above $T_{\text{ice}}$, we employed the MPTRAC model to calculate 24-hour backward trajectories from the point of observation (at $\Delta T_{\text{ice\_min}}$). Fig. 5 displays the fraction of ice PSCs above $T_{\text{ice}}$ along the backward trajectories, ranging from time $t = -24$ to $0$ h. Along the 24-hour backward trajectories, the temperature of most ice PSCs is below $T_{\text{ice}}$. Nevertheless, there is a significant increase in the fraction that occurs within the 6 hours preceding the observations. The fractions of ice PSCs above $T_{\text{ice}}$ increase from 35 % at $t = -6$ h to 52 % at the observation point in the

Antarctic and from 16 % to 26 % in the Arctic. A similar behaviour is observed for ice PSCs with respect to the heterogeneous nucleation threshold ($T = T_{\text{ice}} - 1.5$ K), albeit with a relatively smaller increase.

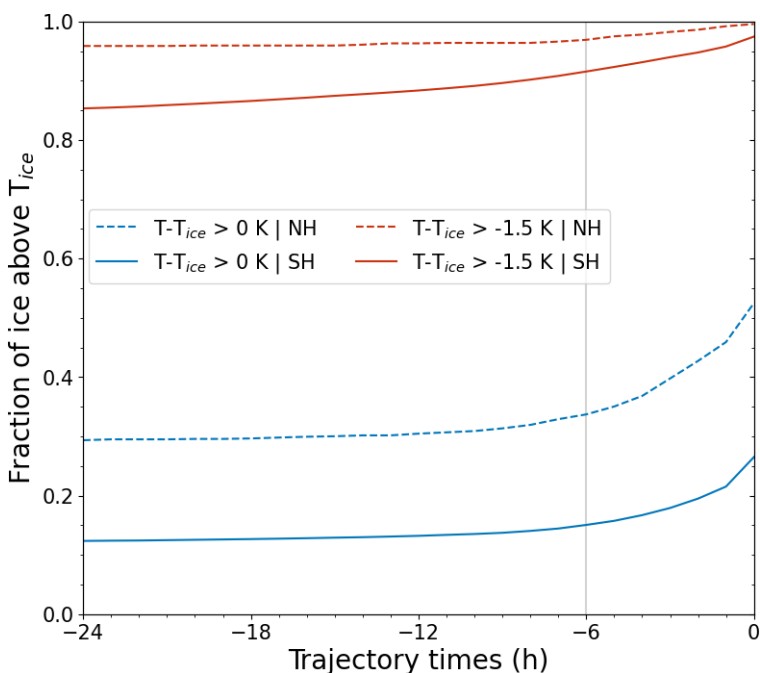

**Figure 5.** Fraction of ice PSCs at temperatures above $T_{\text{ice}}$ and above the heterogeneous nucleation threshold along 24-hour backward trajectories, relative to the total number of ice PSC observations. The solid lines indicate Antarctic data and the dashed lines indicate Arctic data. Blue lines represent temperatures above $T_{\text{ice}}$, while red lines represent temperatures above $T - T_{\text{ice}} > -1.5$ K.

### 3.4 Temperature fluctuations along the backward trajectories

Here, we computed the variance of the cooling rates along the backward trajectories over a 6 h running window, where the variance within each window is assigned to the starting time of that window as it progresses over time. Then, we applied

a variance threshold of $0.9\,\text{K}^2\,\text{h}^{-2}$ to detect the presence of significant temperature fluctuations associated with a gravity wave event. Note that while meteorological reanalyses are often capable of reproducing gravity wave events at the right time and place, especially for mountain waves, the wave amplitudes in the reanalyses are typically damped compared to the real

atmospheric conditions (Schroeder et al., 2009; Jewtoukoff et al., 2015; Hoffmann et al., 2017b). The degree of underestimation depends on the resolution and numerical filters of the forecast model used to generate the reanalysis and on the spectral

characteristics of the gravity waves. Therefore, the simple variance filter method applied here is generally suitable to detect the occurrence of wave events in the atmosphere. However, the detection sensitivity depends critically on the variance threshold. A sensitivity test regarding the choice of the variance threshold is discussed in Sect. 4.4.

An example is presented in Fig. 6 to show how we detected temperature fluctuations along the backward trajectories. The temperature cooling rate and its variance were calculated to identify the temperature fluctuations of the ice PSCs. In this

example, temperature fluctuations were found from $t = -10$ to $0\,\mathrm{h}$, when the temperature cooling rate variance is larger than $0.9\,\mathrm{K^2\,h^{-2}}$, and the temperature is less than $10\,\mathrm{K}$ above $T_{\mathrm{ice}}$. Those detected temperature fluctuations coincide with areas exhibiting high $4.3\,\mu\mathrm{m}$ brightness temperature (BT) variances as retrieved from NASA's Atmospheric Infrared Sounder (AIRS, Fig. 6b), indicating the presence of stratospheric gravity waves (Hoffmann et al., 2013, 2017b). In this particular example, the ERA5 temperature fluctuation over the east coast of Greenland coincides with variations in AIRS brightness temperature,

attributed to the influence of gravity waves. This coincidence suggests a potential connection to the occurrence of ice PSCs.

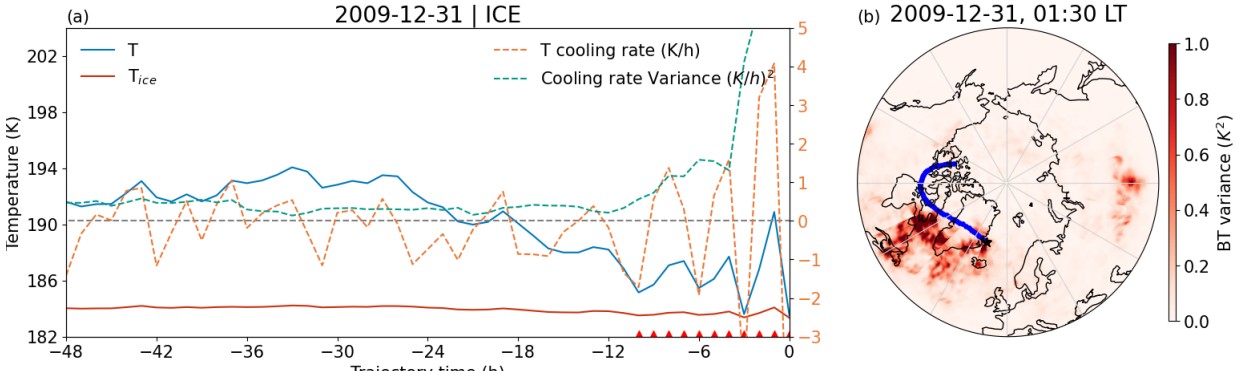

**Figure 6.** Example of detecting temperature fluctuations along backward trajectories. The example shows an ice PSC above $T_{\mathrm{ice}}$ observed on December 31, 2009. a) The temperatures $T$ and $T_{\mathrm{ice}}$ from ERA5 are shown as solid blue and orange lines, and the temperature cooling rate and its variance are dashed orange and green lines. Red triangles indicate the detected temperature fluctuations along the backward trajectory over $t = -10$ to $0\,\mathrm{h}$. b) The brightness temperature (BT) variances detected by AIRS at local time 01:30 are shown as a contour surface in the map. The blue curve shows the backward trajectory, and the black star in the map indicates the location of the observed ice PSC at $\Delta T_{\mathrm{ice\_min}}$.

The cumulative fraction of ice PSCs above $T_{\mathrm{ice}}$ with temperature fluctuations relative to all ice PSCs above $T_{\mathrm{ice}}$ is presented in Fig. 7. Generally, this cumulative fraction increases as we trace backward in time. At the observation points, the mean fractions of ice PSCs above $T_{\mathrm{ice}}$ with temperature fluctuations are 33 % in the Arctic and 9 % in the Antarctic. As we progress to $t = -24\,\mathrm{h}$ (24 hours before the MIPAS observation), approximately 74 % of ice PSCs above $T_{\mathrm{ice}}$ in the Arctic, and about

245 22 % in the Antarctic, could be related to temperature fluctuations (see Table 1). This suggests that the time and location of temperature variations in ERA5 can still be related to the presence of ice PSCs above $T_{\mathrm{ice}}$, especially in the Arctic, even if we consider that gravity wave amplitudes may be underestimated.

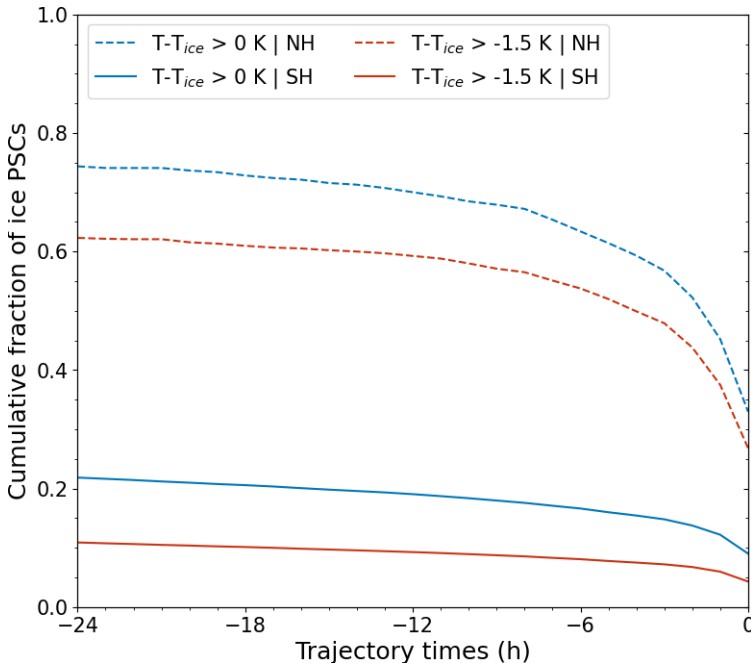

**Figure 7.** The cumulative fraction of ice PSCs above $T_{\text{ice}}$ with temperature fluctuations to all ice PSCs above $T_{\text{ice}}$ over the backward trajectory. Blue lines represent temperatures above $T_{\text{ice}}$, while red lines represent temperatures $T - T_{\text{ice}} > -1.5\,\text{K}$.

Figure 8 presents the spatial distribution of ice PSCs above $T_{\text{ice}}$ with temperature fluctuations at the observation point. The patterns closely resemble the occurrence frequency of ice PSCs above $T_{\text{ice}}$. In the Antarctic, ice PSCs above $T_{\text{ice}}$ with temperature fluctuations are primarily concentrated at and around the Antarctic Peninsula and the Weddell Sea. Notably, two prominent hotspots of ice PSCs above $T_{\text{ice}}$ with temperature fluctuations are situated downwind of the Antarctic Peninsula and Victoria Land. In the Arctic, ice PSCs above $T_{\text{ice}}$ with temperature fluctuations are observed within the longitude range of 60°W to 120°E, encompassing the east coast of Greenland and Northern Scandinavia. In summary, the presence of ice PSCs above $T_{\text{ice}}$ with temperature fluctuations is associated with mountain regions and their downwind areas.

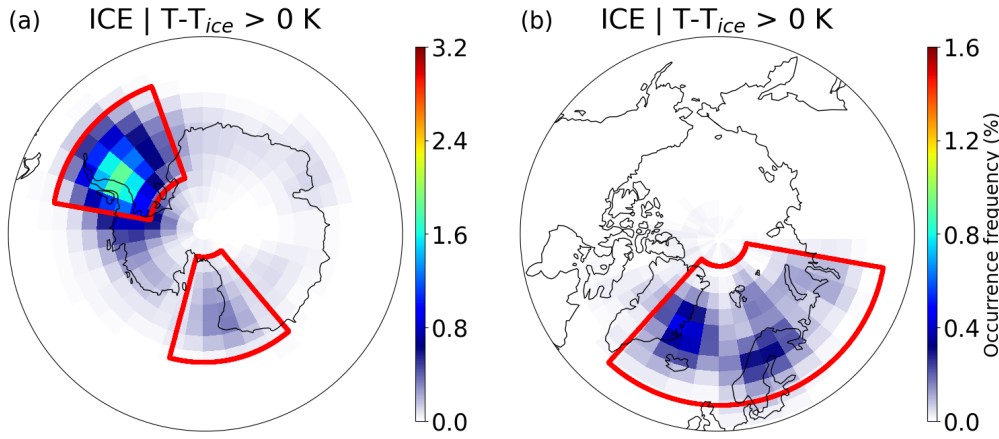

**Figure 8.** Occurrence frequency of ice PSCs above $T_{\text{ice}}$ with temperature fluctuations at the observation point relative to all measurements in the Antarctic (a) and the Arctic (b), respectively. Three specific mountain regions, the Antarctic Peninsula (AP), the Transantarctic Mountains (TM), and the mountain region in the Northern Hemisphere (NH-M), are marked with red boxes.

### 3.5 Simultaneous occurrence of mountain waves and ice PSCs above $T_{\text{ice}}$

To further explore the potential occurrence of ice PSCs above $T_{\text{ice}}$ in connection with orographic waves, we selected two mountain regions in the Antarctic and one in the Arctic (indicated by red boxes in Fig. 8). The defined mountain regions are the Antarctic Peninsula (AP: [$58° - 83°$S, $35° - 80°$W]), the Transantarctic Mountains (TM: [$62° - 85°$S, $165°$W $- 150°$E]) and mountain regions in the Northern Hemisphere (NH-M: [$55° - 85°$N, $45°$W $- 90°$E]). To quantify the potential influence of mountain waves on the occurrence of ice PSCs above $T_{\text{ice}}$, we present the fractions of ice PSCs above $T_{\text{ice}}$ over these specified mountain regions in Table 1, columns $3 - 5$.

Along the trajectories the fraction of ice PSCs above $T_{\text{ice}}$ related to mountain waves decreases as it gets closer to the observation point. At $t = -24$ h, the cumulative fractions of ice PSCs above $T_{\text{ice}}$ related to mountain waves are 9.4 % over the AP and 3.2 % over the TM. However, at the observation point ($t = 0$ h), these fractions are considerably smaller. The fractions related to the Arctic mountain regions are notably higher than those in the Antarctic, reaching 59 % at $t = -24$ h. This difference can be attributed to the higher importance of gravity wave activity on ice PSC occurrence in the Northern Hemisphere, although the size of the selected area also influences the results. In conclusion, ice PSCs above $T_{\text{ice}}$ in the Arctic are more susceptible to the effects of mountain waves.

**Table 1.** Cumulative fraction of ice PSCs related to mountain waves at different backward trajectory times relative to all ice PSCs observed at temperatures above $T_{\mathrm{ice}}$. Mountain regions are the Antarctic Peninsula (AP), the Transantarctic Mountains (TM) and mountain regions in the Northern Hemisphere (NH-M).

| Time | Ice PSCs at $\Delta T_{\mathrm{ice\_min}}$ | | | | |
|------|------|------|------|------|------|
| | SH | NH | AP | TM | NH-M |
| $t = 0\,\mathrm{h}$ | 9.0% | 33.0% | 4.3% | 0.9% | 27.0% |
| $t = -6\,\mathrm{h}$ | 16.6% | 63.4% | 7.6% | 1.6% | 50.3% |
| $t = -12\,\mathrm{h}$ | 19.0% | 70.0% | 8.5% | 2.0% | 55.9% |
| $t = -18\,\mathrm{h}$ | 20.6% | 72.8% | 9.0% | 2.6% | 57.6% |
| $t = -24\,\mathrm{h}$ | 21.8% | 74.4% | 9.4% | 3.2% | 58.7% |

## 4 Discussion

### 4.1 PSC detection in warm environment

In this study, we analyzed the MIPAS-observed ice PSCs and their relation to temperature fluctuations and stratospheric gravity waves. We found that in the Arctic approximately 52 % and in the Antarctic approximately 26 % of the observed ice PSCs are detected at temperatures above $T_{\mathrm{ice}}$ derived from the ERA5 reanalysis. The occurrence of ice PSCs in warm environments have already been reported in previous studies. Based on CALIPSO measurements, Pitts et al. (2018, Fig. 12) found that about 30 % of the ice PSCs in the Arctic and Antarctic are observed above $T_{\mathrm{ice}}$. Spang et al. (2018, Fig. 3) showed that about 20 % of ice PSCs in the Antarctic are observed above $T_{\mathrm{ice}}$ from MIPAS observations at the tangent point over the years 2006 to 2012. Additionally, individual cases of ice PSC detections from MIPAS observations are reported in warm environments with temperatures around or warmer than $T_{\mathrm{NAT}}$, i. e., 5 K or more above $T_{\mathrm{ice}}$, based on the ERA-Interim reanalysis (Hoffmann et al., 2017b, Fig.11). Even if ice PSCs are observed in synoptic-scale warm environments, it is crucial to note that small-scale temperature fluctuations associated with gravity waves can still significantly influence the occurrence of ice PSCs (Dörnbrack et al., 2002; Alexander et al., 2013; Hoffmann et al., 2017b). While global reanalyses or coarse-resolution chemistry-climate models may not fully resolve subgrid-scale temperature fluctuations associated with gravity waves, ERA5 exhibits improved spatiotemporal resolution compared to ERA-Interim. However, ERA5 still tends to underestimate real temperature fluctuations in the polar lower stratosphere. The statistical analysis applied in this study (see previous sections) provides important information on the frequency of these events and once again points out the significance of the discrepancies between observed ice PSCs and temperatures derived from meteorological reanalysis.

### 4.2 Sampling uncertainty in MIPAS for ice PSC detection

To account for the uncertainty in cloud location within MIPAS observations, we present the ice PSCs detected at the tangent point in Fig. 9. Generally, the altitudes of ice PSCs at $\Delta T_{\mathrm{ice\_min}}$ (Fig. 1) are consistently higher than at the tangent point. The

290 highest occurrence frequency of ice PSCs is approximately 4 km higher in the Antarctic at $\Delta T_{\text{ice\_min}}$ than at the tangent point, and around 2 km higher in the Arctic. When comparing ice PSC observations derived from CALIOP (Pitts et al., 2018, Fig. 16 c), occurrence frequencies of ice PSCs derived from MIPAS are relatively higher at the $\Delta T_{\text{ice\_min}}$. This discrepancy may be attributed to the large vertical field of view and coarse vertical sampling resolution of MIPAS. Consequently, cloud top heights of optically thick clouds in MIPAS are probably overestimated, on average by 0.75 km (Sembhi et al., 2012; Griessbach et al.,

2020). Conversely, cloud top heights of optically thin PSCs observed at the tangent point are, on average, underestimated by $0-2$ km and may reach up to 8 km compared to CALIOP (Höpfner et al., 2009).

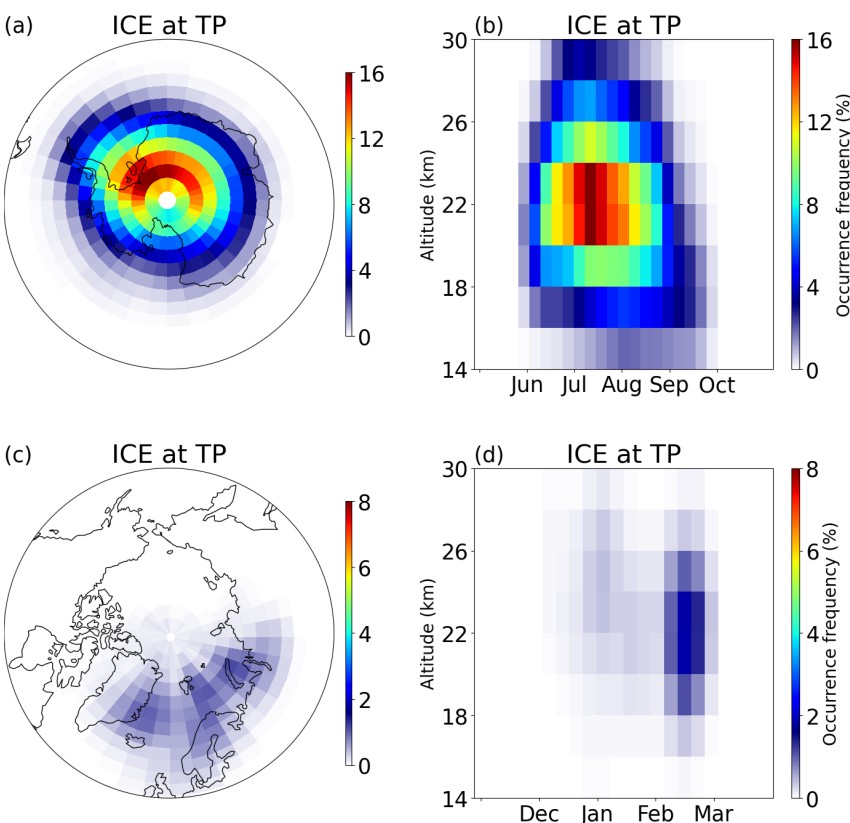

**Figure 9.** Occurrence frequency of ice PSCs (detected PSCs with respect to all measurements) at the tangent point derived from MIPAS observations (2002-2012) for the Antarctic (a, b) and Arctic (c, d). In the horizontal, a), c), the data is gridded on $3° \times 5°$ boxes. In the vertical, b), d), the vertical bin size is 2 km and the data was averaged over 7 days. This figure shows the same as Fig. 1, but for the tangent point.

Furthermore, when considering ice PSC observations as a function of $T - T_{\text{ice}}$, we observe that more ice PSCs are located above $T_{\text{ice}}$ at the tangent point (red lines in Fig. 10) than at $\Delta T_{\text{ice\_min}}$ (gray lines) in both polar regions. The distribution of ice PSCs at $\Delta T_{\text{ice\_min}}$ as a function of $T - T_{\text{ice}}$ is more comparable to the CALIOP observations (Pitts et al., 2018, Fig. 12) and (Tritscher et al., 2021, Fig. 14 a and c) than that at the tangent point. Despite the height discrepancy, the location of $\Delta T_{\text{ice\_min}}$ is more reasonable for the existence of ice PSCs, as MIPAS is more sensitive to optically thin PSCs than CALIOP (Sembhi et al., 2012; Griessbach et al., 2020). The method to detect PSC locations in MIPAS observations, based on $\Delta T_{\text{ice\_min}}$, complements the conventional tangent point approach and aids in determining cloud positions within this study.

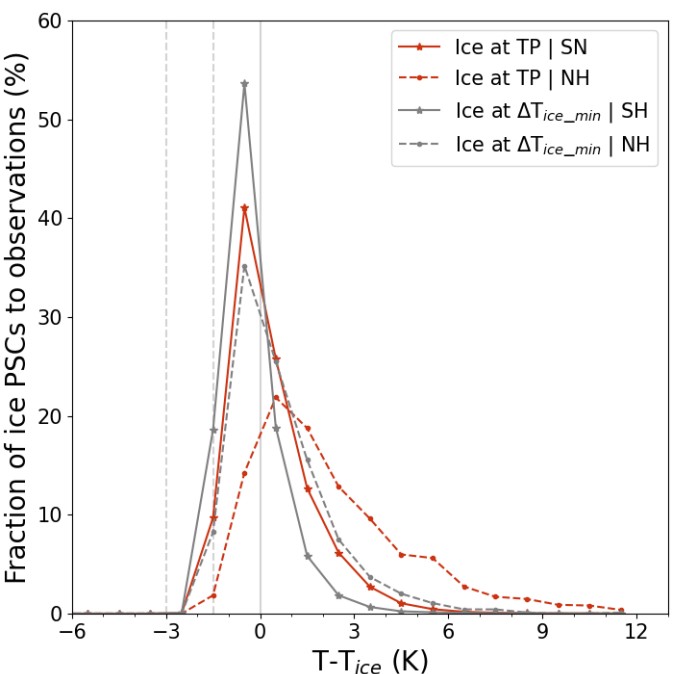

**Figure 10.** Occurrence frequency of ice PSCs as a function of the temperature difference $(T - T_{\text{ice}})$. The red lines indicate the distribution at the tangent point. For comparison, the grey lines for the observations at $\Delta T_{\text{ice\_min}}$ are the same as in Fig. 2. The solid lies indicate data from the Antarctic and the dashed lines indicate data from the Arctic.

### 4.3 Temperature and $T_{\text{ice}}$ uncertainties in the ERA5 reanalysis

In this study, ambient temperatures of MIPAS observation points are interpolated from the ERA5 reanalysis, which offers significantly improved spatial and temporal resolution compared to ERA-Interim (Hoffmann et al., 2019; Hersbach et al., 2020). Nevertheless, it is essential to acknowledge that the ERA5 reanalysis provides a global mean temperature estimate with an uncertainty of approximately 0.2 K compared to radiosonde measurements in the low and middle stratosphere (Simmons et al., 2020). Additionally, it is crucial to note that the ERA5 reanalysis may not fully resolve temperature perturbations associated with various factors such as convective updrafts, gravity waves, and other meso- to synoptic-scale features, as discussed in

Hoffmann et al. (2019). In particular, wave amplitudes are often underestimated. Consequently, there is a possibility that some temperature fluctuations are not resolved in ERA5 as discussed in Sect. 3.4.

In addition to ambient temperature, the calculation of $T_{ice}$ utilizes water vapor data ($H_2O$) from ERA5 and pressure, applying the equation proposed by Marti and Mauersberger (1993). However, the uncertainties in water vapor data from ERA5 remain unclear. Different methods for calculating $T_{ice}$ may introduce additional uncertainty in identifying ice PSCs above $T_{ice}$. Figure 11 presents a sensitivity analysis of how $T_{ice}$ varies with water vapor content and pressure depending on the calculation method. Taking the example of a pressure level equal to 50 hPa (about 20 km of height), the $T_{ice}$ uncertainty is less than 2 K when the water vapor content ranges between typical stratospheric values of 2 ppm and 5 ppm. Different calculation methods for $T_{ice}$ result in negligible uncertainty, even though $T_{ice}$ calculated by Marti and Mauersberger (1993) is slightly warmer than following more recent methods (Murphy and Koop, 2005; WMO, 2008).

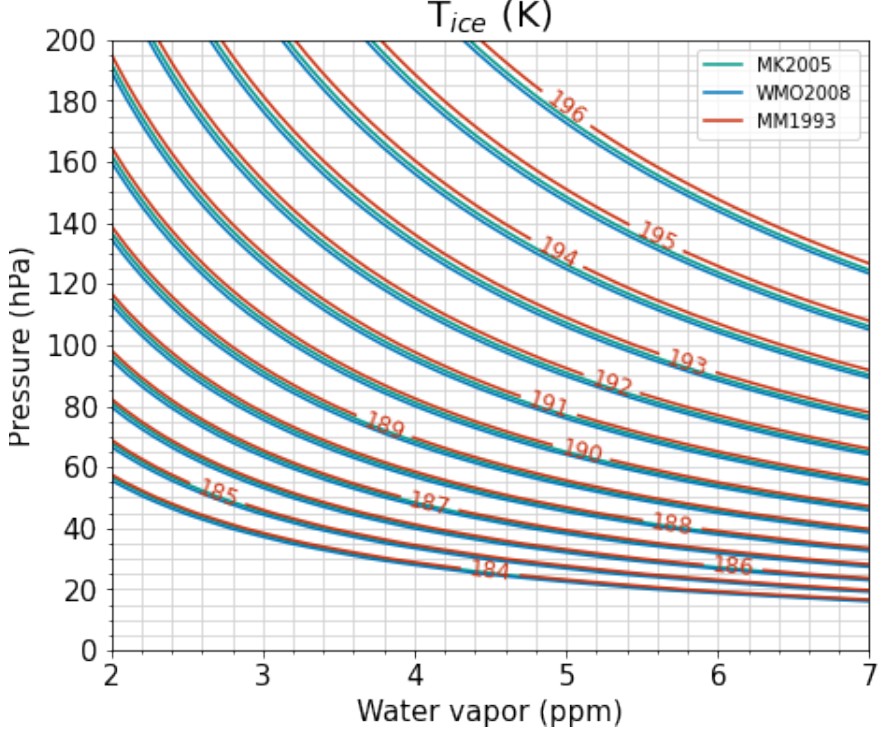

**Figure 11.** Sensitivity of $T_{ice}$ as a function of water vapor content to various calculation methods. MM1993, MK2005 and WMO2008 are methods proposed by Marti and Mauersberger (1993), Murphy and Koop (2005) and WMO (2008), respectively.

## 4.4 Temperature fluctuation uncertainty

The uncertainty in the selection of the threshold for the variance of the temperature cooling rate to detect temperature fluctuation events is illustrated in Fig. 12, where different thresholds for temperature cooling rate variance ($T_{crv}$) are examined, ranging from $0.6\,\mathrm{K^2\,h^{-2}}$ to $1.2\,\mathrm{K^2\,h^{-2}}$ in increments of $0.1\,\mathrm{K^2\,h^{-2}}$. As $T_{crv}$ increases, fewer temperature fluctuations are detected. For

instance, the highest fraction of ice PSCs above $T_{ice}$ with temperature fluctuations is observed at $T_{crv} = 0.6\,\mathrm{K^2\,h^{-2}}$, where the fraction is approximately 30 % at $t = -24$ h. Conversely, the smallest fraction is found at $T_{crv} = 1.2\,\mathrm{K^2\,h^{-2}}$, with a fraction of 18 % at $t = -24$ h. However, the uncertainty in the choice of $T_{crv}$ decreases as the time approaches the observation point. At the MIPAS observation point, the uncertainty of $T_{crv}$ is within 1 percentage point in the Antarctic and 2 percentage point in the Arctic, with increments of $0.1\,\mathrm{K^2\,h^{-2}}$.

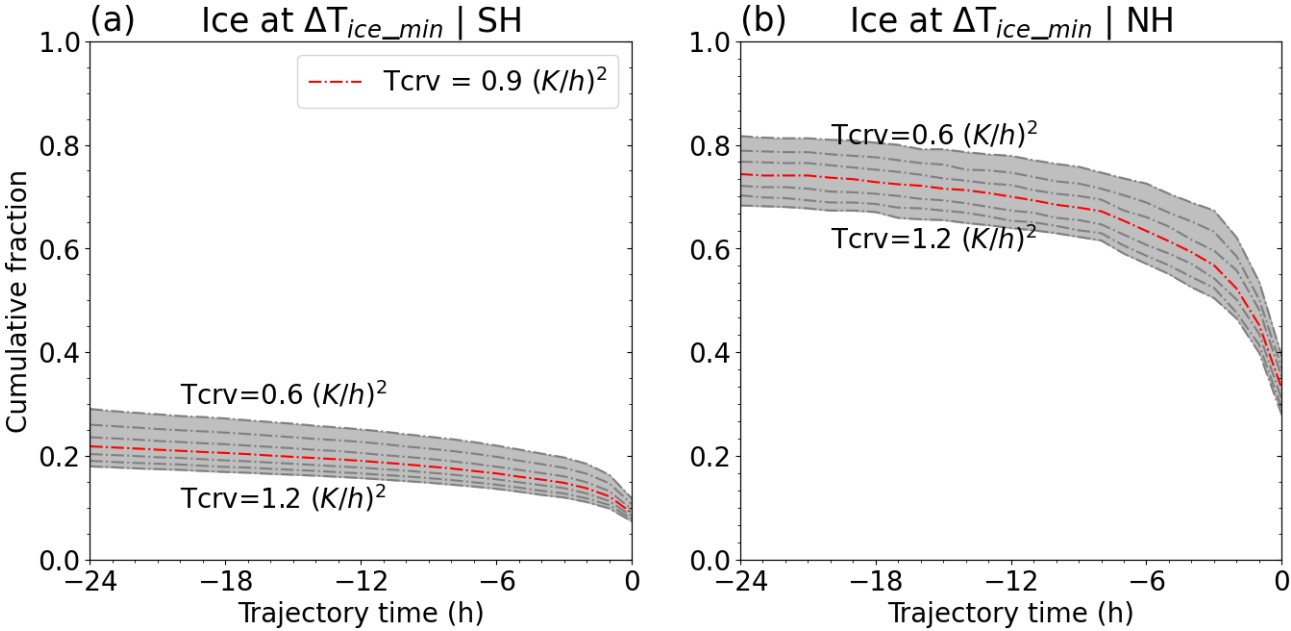

**Figure 12.** Cumulative fraction of ice PSCs above $T_{ice}$ with different temperature cooling rate variance thresholds ($T_{crv}$) for temperature fluctuation detection. $T_{crv} = 0.9, ..., 1.2\,\mathrm{K^2\,h^{-2}}$, the red dashed line is $T_{crv} = 0.9\,\mathrm{K^2\,h^{-2}}$ as selected for the statistical assessment.

The fractions of all ice PSCs above $T_{ice}$ passing through the defined mountain regions along the backward trajectory are summarized in Table 2. These values exhibit a substantial increase compared to the fractions of ice PSCs above $T_{ice}$ with temperature fluctuations in mountain regions (Table 1). For example, at the MIPAS observation point in the AP region, the fraction of ice PSCs above $T_{ice}$ is 18.2 %, whereas the fraction with temperature fluctuations is only 4.3 % as shown in Table 1. Comparing the values in Table 1 and Table 2, we find that although the majority of ice PSCs above $T_{ice}$ pass through specified 335 mountain regions, only a limited number of them exhibit temperature fluctuations (Table 1). One possible reason for this discrepancy is that the temperature fluctuations detected in our study may be underestimated due to the selected $T_{crv}$ threshold. Once again, unresolved temperature fluctuations in ERA5 reanalysis may contribute to these differences. Other reasons, such as the formation mechanism of gravity waves (Dörnbrack et al., 2001), could also affect the results, but taking these into account is beyond the scope of this study.

**Table 2.** Fraction of ice PSCs above $T_{\text{ice}}$ passing through specified mountain regions at different backward trajectory times.

| Time | Ice PSCs at $\Delta T_{\text{ice\_min}}$ | | |
|---|---|---|---|
| | AP | TM | NH-M |
| $t = 0\,\text{h}$ | 18.2% | 8.2% | 73.9% |
| $t = -6\,\text{h}$ | 20.7% | 11.7% | 79.6% |
| $t = -12\,\text{h}$ | 22.6% | 16.4% | 83.7% |
| $t = -18\,\text{h}$ | 24.0% | 22.7% | 83.9% |
| $t = -24\,\text{h}$ | 25.3% | 30.0% | 84.1% |

## 5 Conclusions

This study examines a decade-long (2002 – 2012) record of ice PSCs derived from MIPAS/Envisat measurements. The points with the smallest temperature difference ($\Delta T_{\text{ice\_min}}$) between the frost point temperature ($T_{\text{ice}}$) and the environmental temperature along the line of sight have been proposed and shown to provide a better estimate of the location of ice PSC observation from MIPAS. The temperature at the ice PSC observations is analyzed based on the ERA5 reanalysis. Following this, we investigated the temperature history of the ice PSCs detected above $T_{\text{ice}}$ at the observation points along 24-hour backward trajectories.

In the MIPAS observations, ice PSCs are mostly observed in the longitude range of 90°W to 90°E in the Antarctic with peak values over 16 %, and between 60°W to 120°E in the Arctic during midwinter with peak values of about 2 %. Ice PSCs at $\Delta T_{\text{ice\_min}}$ are mostly detected in the altitude range of 22 km to 26 km, which is about $2 - 4$ km above the tangent point.

The occurrence frequencies of ice PSCs as a function of temperature difference to $T_{\text{ice}}$ during the period $2002 - 2012$ show that approximately 52 % and 26 % of the ice PSCs in the Arctic and Antarctic, respectively, are detected at temperatures above the local $T_{\text{ice}}$ derived from temperature and water vapor data from ERA5. In the Antarctic, ice PSCs above $T_{\text{ice}}$ are predominantly located around the Antarctic Peninsula and its downwind direction, notably towards the Weddell Sea. In the Arctic, ice PSCs above $T_{\text{ice}}$ are distributed across mountain areas such as East Greenland and northern Scandinavia, along with their respective downwind directions. The ice PSCs above $T_{\text{ice}}$ are mainly concentrated in deep winter and descend in altitude during the course of the winter.

24-hour backward trajectories were calculated by using the MPTRAC model from the ice PSC observations at $\Delta T_{\text{ice\_min}}$. The most significant change of the fraction of ice PSCs above $T_{\text{ice}}$ occurs within the 6 hours preceding the observations, in which the fractions increase from 35 % at $t = -6$ h to 52 % at the observation point in the Antarctic and from 16 % to 26 % in the Arctic.

Furthermore, temperature fluctuations along the backward trajectories were identified by the temperature cooling rate and its variance. At the observation point, the mean fractions of ice PSCs above $T_{\text{ice}}$ with temperature fluctuations are 33 % in the Arctic and 9 % in the Antarctic. 24 hours before the MIPAS observation the fraction of ice PSCs that have experienced temperature fluctuations increased to approximately 74 % in the Arctic and about 22 % in the Antarctic. Despite being underestimated in

their magnitude, the temperature fluctuations in ERA5 have a significant coincident occurrence with the presence of ice PSCs above $T_{ice}$, particularly in the Arctic. The ice PSCs above $T_{ice}$ with temperature fluctuations along the backward trajectories are primarily concentrated over the Antarctic Peninsula and the Weddell Sea in the Antarctic and encompass the east coast of Greenland and Northern Scandinavia in the Arctic, which are known hotspots of mountain wave activity.

Across specified mountain regions, the fractions of ice PSCs above $T_{ice}$ related to mountain waves are 9.4 % over the AP and 3.2 % over the TM for observations at $\Delta T_{ice\_min}$ at $t = -24\,h$. However, at the observation point ($t = 0\,h$), these fractions are considerably smaller. The fractions in the Arctic mountain regions, reaching 59 %, are notably higher than those in the Antarctic. This difference can be attributed to the larger role of gravity wave activity in the occurrence of ice PSCs in the Northern Hemisphere. This is due to the generally higher temperatures in this hemisphere, which often require mountain wave-induced temperature fluctuations to initiate ice PSC formation. However, the larger size of the selected region may also contribute to the hemispheric differences.

Our results are subject to several uncertainties. The uncertainties of temperature and water vapor in ERA5 data impact the identification of ice PSCs. We may also miss or underestimate many small-scale temperature fluctuations along the backward trajectories, which are not fully resolved in the ERA5 data, and the choice of the temperature cooling rate variance threshold for detecting gravity wave events has also an impact on the results. Furthermore, substantial differences in cloud heights exist between MIPAS observations assigned to $\Delta T_{ice\_min}$ or assigned to the tangent point. Also, MIPAS measurements are integrated along the long limb path, but temperatures retrieved from ERA5 are for a spatial resolution of 31 km ($\Delta T_{ice\_min}$ or tangent point), which produces an uncertainty for identifying the distribution of ice PSCs relative to $T_{ice}$. Investigating the source of the discrepancies between ice PSC observations and warm temperatures is pertinent for understanding the formation of ice PSCs, as they require temperatures that are significantly below $T_{ice}$.

*Code and data availability.* The MIPAS operational data are provided by the European Space Agency. The MIPAS PSC data product version 1.2 used in this study is accessible at https://datapub.fz-juelich.de/slcs/mipas/psc/index.html (Spang, 2020). The ERA5 data are provided by the the European Centre for Medium-Range Weather Forecasts, see https://www.ecmwf.int/en/forecasts/datasets (Hersbach et al., 2020). The MPTRAC model used in this study has been archived on Zenodo (Hoffmann et al., 2021, https://zenodo.org/records/5714528).

*Author contributions.* The conceptualization was conducted by L.Z., R.S., L.H., S.G., F.K., R.M., and I.T. R.S. provided the PSCs data repository retrieved from MIPAS and L.H. provided the MPTRAC model, while L.Z. processed the data and compiled all results. L.Z. wrote the manuscript with contributions from all co-authors.

*Competing interests.* At least one of the (co-)authors is a member of the editorial board of Atmospheric Chemistry and Physics. Otherwise the authors declare no competing interests.

*Acknowledgements.* This research was supported by the Helmholtz Association of German Research Centres (HGF) through the Joint Laboratory for Exascale Earth System Modeling (JL-ExaESM) at Forschungszentrum Jülich. The authors are grateful to the Jülich Supercomputing Center for providing computing time and storage resources on the JUWELS supercomputer.

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
