# Peer review of "Impact of mountain-wave-induced temperature fluctuations on the occurrence of polar stratospheric ice clouds: A statistical analysis based on MIPAS observations and ERA5 data"

_EGUsphere, 2024_

## Author Response (AR1)

**Reply to Referee 1**

**We thank Referee 1 for the constructive, helpful criticism and the suggestion for revision. We have thoroughly revised the manuscript based on the comments given by the referees. A detailed point-by-point response to the comments by Referee 1 is given below.**

*In this article, the authors propose a way to evaluate the fraction of ice Polar Stratospheric Clouds (PSC) affected by short-lived small-scale temperature fluctuations generated by gravity waves, using a combination of MIPAS PSC detections and reanalyses temperatures. They support their analysis by the use of backward trajectories to document the temperature history of air masses. Their results bring new quantified information on the relationship between ice PSC and gravity wave activity over both poles.*

*The article describes the methodology and results in a very clear way and is extremely readable. The proposed methodology is sound, the results are consistent with what is known about PSCs and gravity waves, while bringing new information that completes what we know in a useful manner. The authors clearly have a good handle on the data from MIPAS and reanalyses, which are very well described. References abound, the bibliography is complete and well selected. The figures are well designed and convey their message with clarity. I support the publication of this article in ACP, once the few remarks I have below have been addressed.*

**We would like to thank the referee for the encouraging statement.**

*Major comment*
*My main comment is a suggestion to the authors to clarify their intent, make some of their assumptions explicit, and better relate the approach they've devised with the scientific questions that are raised. As currently written, it is a bit hard to understand why the authors are doing what they are doing.*

**We agree with the referee that the aspects mentioned here and the major objectives of the study need to be introduced more clearly. Following the comments and suggestions, we carefully revised and extended the abstract and the introduction to clarify.**

*For instance, the abstract begins with "Small-scale temperature fluctuations can play a crucial role in the occurrence of ice clouds". This may be true, but rather vague. What small-scale temperature fluctuations are we talking about? Are we talking about temperature fluctuations created by mountain/orographic waves? Are "ice clouds" meant to be ice PSC? This first sentence is the only one that suggests what questions the authors are interested in. The rest of the abstract describes the methodology and results, but fails in my opinion to relate them to the scientific questions raised by the first sentence. After the first sentence, the abstract goes on explaining that the study is analyzing PSC occurrences from MIPAS, using the smallest difference temperature etc. How are all these steps related to better understanding the crucial role the small-scale temperature fluctuations play on the occurrence of ice clouds? Please fill in the gaps.*

**We revised the first sentence of the abstract to specify that we are referring to temperature fluctuations caused by gravity waves, including those generated by mountain/orographic waves. We also clarified that the "ice clouds" referred to are indeed ice polar stratospheric clouds (PSCs) and added some sentences to relate our study to the scientific question.**

*I was hoping the introduction would clarify these points, by describing how the methodological approach chosen by the authors is appropriate to answer the questions raised by that first sentence, but in my opinion it suffers from the same problem as the abstract. The first 3 paragraphs do a good job explaining why temperature perturbations generated by gravity waves affect the formation of PSC. The last paragraph dives straight in a summary of the selected methodology, but does not relate this methodology to the questions raised in the previous paragraphs, or explain what insights it will bring. I feel there is a missing paragraph before this one that should explain why MIPAS detecting a PSC, when ERA5 says the coldest synoptic temperatures are too warm to support PSC formation, is a sign of gravity wave activity. I have felt this as an implicit assumption throughout the whole text – please make it explicit. I understand that the authors perhaps want to be careful and avoid stating it for some reason, but not making it explicit makes it hard to appreciate what the authors' work brings to the subject under study. In addition to the abstract and the introduction, the link between the authors' approach and gravity wave activity needs to be made clearer in section 4 and 5 also.*

**We revised the introduction to clearly state that a major objective of the study is to assess the effects of unresolved or only poorly-resolved temperature fluctuations due to gravity waves on PSC formation. Please also see our replies to the specific comments below.**

*Minor comments*

- *Title: the current title is very generic. Please at least refer to gravity waves in it.*

  **Done. We have changed the title to "Impact of mountain-wave-induced temperature fluctuations on the occurrence of polar stratospheric ice clouds: A statistical analysis based on MIPAS observations and ERA5 data"**

- *L. 1: "a decade" please state which years we are talking about here (2002-2012 ?)*

  **Yes, the time period considered is 2002-2012. We have added this.**

- *L. 18: "The surfaces of PSCs" I don't think we can say that clouds have surfaces. Maybe "the surface of PSC particles"*

  **Fixed**

- *L. 46-47: This paragraph begins by explaining the primary focus of the study is to investigate the occurrence of ice PSC observed by MIPAS and characterized by temperatures above the ice existence threshold. Please explain why, when interested in the relationship between ice PSC and gravity waves (as the beginning of the section suggests), it would be a good idea to do that (see major comment).*

  **Following the major comment and comments of other referees, we added a new paragraph to the introduction that explains the physical processes of how gravity wave activity leads to PSC formation in more detail: "Atmospheric gravity waves are oscillations in the Earth's atmosphere caused by buoyancy and gravity acting on air parcels displaced from their equilibrium positions. They can significantly affect local pressure, temperature, winds, and other meteorological variables. Gravity waves play a critical role in the transfer of energy and momentum between different layers of the atmosphere, i.e., they can significantly affect the temperature structure and general circulation of the atmosphere (Fritts and Alexander, 2003;**

Alexander et al., 2010). Gravity waves typically originate from sources such as mountain ranges, where air flow is disturbed and lifted; thunderstorms, where convective activity is generated; and frontal systems, where air masses are disturbed from geostrophic equilibrium. Understanding gravity waves is essential for accurate weather forecasting and broader atmospheric dynamics. Properly representing the effects of gravity waves in general circulation and chemical transport models is challenging because coarse-grid global models typically lack the spatial resolution needed to resolve the small-scale perturbations of gravity waves. In this study, we are particularly interested in atmospheric gravity waves in the polar winter lower stratosphere, because temperature perturbations due to gravity waves can trigger the development of PSCs in the cold phase of these waves, even in cases where background or synoptic-scale temperatures are higher than their formation thresholds (Carslaw et al., 1998; Rivière et al., 2000; Dörnbrack et al., 2020; Orr et al., 2020). Missing these waves in coarse-resolution global chemistry-climate simulations or reanalysis-driven chemistry-transport simulations will affect the PSC representation."

- *L. 76-85: Here you explain your alternative to using the tangent point for identifying the location of a PSC that MIPAS has detected along its line of sight. Have you tried to investigate the spatial distribution of temperatures along the line of sight? Is it frequent for cold temperatures to be spatially spread along the line of sight far from the point identified as the coldest? Or can you have two locations along the line of sight with similar coldest temperatures? This would mean the PSC location is affected by strong uncertainties. In other words, how well defined spatially is the $\Delta$Tice_min ?*

Since the real location of a PSC along the MIPAS line of sight remains unknown as ground truth, it is difficult to quantify the uncertainty and improvements in locating the PSC detections with different methods. Assigning the detection to the tangent point, as in previous studies, is a heuristic choice. Similarly, assigning the detection to the location of the minimum of the temperature difference between Tice and T along the line of sight ($\Delta$Tice_min) is also a heuristic but likely more plausible choice.

In Sect. 4.2 of the paper, we discuss the sampling uncertainty of MIPAS for the ice PSC detections. In particular, we evaluated how the choice of different methods of locating the PSCs (tangent point versus $\Delta$Tice_min) affects the spatial and temporal distributions of the observed occurrence frequencies (compare Figs. 1 and 9, respectively). We found a larger sensitivity to the height difference than to the horizontal distance of the tangent point and the location of the $\Delta$Tice_min minimum, which we attribute to the fact that temperature changes significantly with height but not so much horizontally. In the statistical comparison with $3° \times 5°$ horizontal grid boxes conducted here, the horizontal distribution of ice PSCs is found to be essentially identical for both methods.

Although we cannot quantify the uncertainty of the individual locations of the MIPAS PSC detections due to the lack of ground truth information, we performed an additional statistical analysis to determine the horizontal and vertical distances between the tangent point and the temperature difference minimum along the line of sight (see Fig. 1 in this reply). This analysis shows that half of the time the horizontal distances are below $\pm$ 200 km and the vertical distances are below 4 km.

**This is consistent with the discussion in Sect. 4.2 of the paper**

[Figure]

Figure 1: Histograms of the horizontal (left) and vertical (right) distances between the tangent point and the minimum of the temperature difference between $T_{ice}$ and $T$ along the line of sight. The statistics cover MIPAS observations during Southern Hemisphere polar winter in the years 2003 to 2011.

- *L. 101, 103: References to Hoffmann et al. 2017b and a are missing parentheses. Also please fix the citations (a should be cited before b).*

  **Thanks for pointing this out. The missing parentheses have been added. We agree that a should cited before b. However, this order is done by bibtex automatically (based on the names of the co-authors) and we have not found a solution on how to change this. We hope that the Copernicus Editorial Support Team can fix this during the typesetting/copyediting.**

- *Section 2.2: Please explain why you use both ERA-Interim \*and\* ERA5. Spatial/temporal resolutions appear similar, so why isn't it possible to pick just one reanalysis dataset? If using two datasets is mandatory, could you talk about how potential disagreements in polar stratospheric temperatures from both datasets would affect your results?*

  **Thank you for pointing this out. After rechecking the data, we can confirm that only ERA5 was used in our study. The PSC dataset we use here is the one described in Spang et al. (2018). In the original dataset provided with the paper, temperature information e.g. $T_{ice}$ is based on ERA-interim temperatures. However, updates of this MIPAS PSC dataset including the additional information of temperatures along the line of sight, were provided with ERA5 temperatures. Our original manuscript was based on one of the older versions, and when we updated the MIPAS PSC dataset, we accidentally missed updating the text in the manuscript. We apologize for this oversight and appreciate that this mistake was discovered through the referees' questions. The manuscript has now been revised accordingly. We added the specific version of the MIPAS PSC data product used in this study in the code and data availability section of the manuscript.**

- *Section 2.2: I was under the impression that Hoffmann et al 2017 (the one about Concordiasi) suggested that ERA-Interim suffers from a zonally increasing warm bias in the south pole. Is that affecting your results in any way? Is this bias corrected in ERA5?*

  **We have updated the manuscript that only ERA5 was used in our study. The study of Hoffmann et al. (2017) was focusing on comparisons of polar stratospheric temperatures with Concordiasi super-pressure balloon observations. Most of the balloon measurements were conducted at specific altitudes of 17–18.5 km and latitudes of 60–85°S. Biases between reanalyses and observations generally differ with latitude, height and over the season.**

- *L. 116: what does it mean for the Lagrangian transport to be significantly impacted? Is it better, worse, something else? Does this suggest that using ERA5 (instead of ERA-Interim) for locating and quantifying the $\Delta Tice\_min$ would lead to different results?*

  **To clarify, we rephrased: "The evaluation of MPTRAC trajectory calculations was assessed by using different meteorological reanalyses (Hoffmann et al., 2017; Rößler et al., 2018; Hoffmann et al., 2019). It was found that Lagrangian transport simulations are significantly improved by the ERA5 data compared to ERA-Interim data (Hoffmann et al., 2019). For instance, it was found that there is better conservation of potential temperature along the ERA5 trajectories than the ERA-Interim trajectories in the lower stratosphere."**

- *L. 119: Here you state that your aim is to "conduct a statistical analysis of ice PSCs where the temperature at the MIPAS observation is above the frost point temperature". Again,*

*in my opinion you have not made clear enough why you might want to do that (see main comment). Please make explicit the link with gravity wave activity.*

**To make a clearer link to the main objectives of the study, we rephrased this sentence to: "In order to assess how temperature fluctuations due to gravity waves affect PSC formation using the MIPAS measurements and ERA5 data, we first need to identify ice PSC observations where the temperature at the location of the MIPAS observation is above the frost point temperature (Tice)."**

- *L. 135: After lines 76-85, this is the second time you explain your alternative to the tangent point for PSC detection. You explain it again on lines 240-242. Please try to limit these explanations.*

  **We removed duplicates on lines 135 and 240-242.**

- *L. 139: "highest occurrence frequency": over which time scales? 2% is not a lot.*

  **The time scale has been added. Yes, 2% means the ice PSCs are not so often detected in the Arctic. The sentence has been rephrased as follows: "The spatial and vertical distribution of the averaged occurrence frequency of ice PSCs at $\triangle$Tice_min from MIPAS measurements over the time period $2002 - 2012$ is presented in Fig. 1."**

- *L. 146: Here you describe that most $\triangle$Tice_min are found below the frost point. Reading this confused me at first, since on lines 119 you explain that your focus is on points that are above the frost point. I think you could limit reader confusion that making it clearer from the start that you expect $\triangle$Tice_min above the frost point to be rare, and that you take them as the sign of small-scale temperature perturbations from gravity waves, i.e. by clarifying why you are doing what you are doing (see main comment).*

  **Thanks for the comment. We improved the text by better explaining what we have done and removed sentences that could be misleading.**

- *L. 148-149: The discussion about ice PSC particles nucleating at temperatures colder than Tice mirrors the one found on lines 123-125, but with fewer details. Please find a way to combine both discussions. I think the discussion should happen in Section 3.1, but with the details found in lines 123-125.*

  **Thanks, we have consolidated lines 123-125 and 148-149 in Section 3.1.**

- *L. 151: "T-Tice-3K" I'm guessing here you mean Tice-3K*

  **Yes, indeed. This has been corrected.**

- *L. 175: "Fig. 5 displays the fraction of ice PSCs above Tice". I did not understand this part. As I understand, MPTRAC calculates backward trajectories starting from the point where a PSC was located according to MIPAS observations $+ \triangle$Tice_min from ERA5. MPTRAC provides the evolution of temperature and other parameters along the backtrajectory, but has no way to know when a PSC was present in the air mass that it is tracking, and does not provide that information. Are you assuming that the air mass where a PSC was detected at t=0 contains a PSC over the previous 24 h through the entire backtrajectory? Please clarify my misunderstanding. (also please fix related wording line 309)*

  **You are correct, we don't know when a PSC was present. We used only the evolution of temperature along the back trajectory from the MPTRAC model. We have**

corrected the misleading wording. "24-hour backward trajectories are calculated by using the MPTRAC model from the point of ice PSCs observation at $\triangle$Tice_min."

- *L. 176: "(t) t=-24 (h)" – do you mean t=-24h ?*

  **Yes. This has been corrected.**

- *L. 177-178: It appears strange to me to say the fraction decreases by going from t=0 to t=-6h, ie going backwards in time. It would appear more natural to say the fraction increases from t=-6h to t=0.*

  **We have revised these lines." The fractions of ice PSCs above Tice increase by about 17 percentage points (pp) and 10 pp in the Arctic and Antarctic, respectively, from $t = -6$ h to the observation point at $\triangle$Tice_min ."**

- *L. 180: "6h before the observation, temperatures of most ice PSC... are below Tice". If I read Fig. 5 correctly, it looks to me like temperatures of most ice PSC are below Tice at all times.*

  **Yes, you are correct. We have revised this part. The text reads now: "Over 24-hour backward trajectories, temperatures of most ice PSCs are below Tice. Nevertheless, there is a significant increase of the fraction that occurs within the 6 hours preceding the observations."**

- *Section 3.4: If I understand correctly, your main hypothesis is that the influence of gravity waves on PSC can be inferred by $\triangle$Tice_min being positive, as it means that the reanalysis temperatures are failing to capture small-scale temperature variations due to GW. In this section, however, you look for short-scale temperature variations within the reanalysis as indicators for the influence of gravity waves. There seems to be a contradiction – either GW influence is captured in reanalyses, or it isn't. Please clarify my misunderstanding. Are you expecting ERA5 (used by backtrajectories) to better capture the gravity wave influence on temperatures than ERA-Interim (used for $\triangle$Tice_min )? Why?*

  **We added the following paragraph to clarify: "Here, we computed the variance of the cooling rates along the backward trajectories over a 6 h running window and applied a variance threshold of $0.9\,\mathrm{K}^2\,\mathrm{h}^{-2}$ to detect the presence of significant temperature fluctuations associated with a gravity wave event. Note that while meteorological reanalyses are often capable of reproducing gravity wave events at the right time and place, especially for mountain waves, the wave amplitudes in the reanalyses are typically damped compared to the real atmospheric conditions (Schroeder et al., 2009; Jewtoukoff et al., 2015; Hoffmann et al., 2017). The degree of underestimation depends on the resolution and numerical filters of the forecast model used to generate the reanalysis and on the spectral characteristics of the gravity waves. Therefore, the simple variance filter method applied here is generally suitable to detect the occurrence of wave events in the atmosphere. However, the detection sensitivity depends critically on the variance threshold. A sensitivity test regarding the choice of the variance threshold is discussed in Sect. 4.4."**

- *L. 209-210: "This observation strongly suggests a correlation with orographic waves with ice PSCs above Tice". This reads strange to me. We \*know\* that orographic waves trigger the formation of ice PSCs. Unless I'm mistaken, this is actually the (unstated)*

*reason why you look for positive $\Delta Tice\_min$ – because they are the sign of short-scaled temperature perturbations, that are not well captured by ERA-Interim, and are generated by gravity waves. It looks like you are trying here to avoid stating that we already know that orographic waves trigger PSC formation. Please be explicit about your assumptions.*

**We would like to avoid giving any wrong impression that this would be a new finding of this study. We therefore deleted this sentence. Similarly, at lines 317 to 320, we rephrased: "The ice PSCs above Tice with temperature fluctuations along the backward trajectories are primarily concentrated over the Antarctic Peninsula and the Weddell Sea in the Antarctic and encompass the east coast of Greenland and Northern Scandinavia in the Arctic, which are known hotspots of mountain wave activity."**

- *L. 218-223: I am very confused by this paragraph. I understand your results find that the fraction of $T > Tice$ related to mountain waves are higher in the Arctic than in the Antarctic (l. 220-223), no problem there. But, unless I'm mistaken, your main point is that the fraction of $T > Tice$ related to mountain waves decreases as it gets closer to the observation point. This is what you open your paragraph with, and the main result from Table 1. And yet you make no attempt to explain this decrease. Why is this fraction decreasing as we get closer to observation point? I personally find this result perplexing.*

  **We have revised this paragraph. Table 1 shows the cumulative fraction of ice PSCs above Tice over certain regions, demonstrating that the different timescales of backward trajectories affect the results. As longer backward trajectories are included, more ice PSCs are found to be associated with mountain waves.**

- *L. 230: double parentheses*

  **The obsolete parentheses have been deleted.**

- *L. 226-238: This paragraph is the closest you get to state your assumption that PSCs observed at $\Delta Tice\_min < 0$ are due to small-scale temperature fluctuations from gravity waves that brought temperatures below Tice in a way that is not captured by ERA-interim reanalyses. Still, you need to make this reasoning explicit. Also: if the main hypothesis is that ERA-Interim misses temperature fluctuations that generate PSCs, why should we trust the location of the $\Delta Tice\_min$ according to ERA-Interim? The PSC detected by MIPAS might be somewhere else along the line of sight, in a place more affected by gravity waves (which are not well captured by ERA-Interim). Please address this somehow.*

  **Following several review comments, we have revised the abstract, motivation, and conclusions sections of the paper to clarify the main objective and purpose of this study. In this paragraph, our new results on MIPAS ice PSC detections related to temperature variations due to gravity waves are placed in the context of existing results and studies in the literature. We have slightly revised the paragraph to clarify its purpose.**

- *L. 317: "with temperature fluctuations at the observation point" – as I understand it, the observation point is fixed in time and space. How can there be temperature fluctuations are the observation point then?*

  **We rephrased this to clarify: "The ice PSCs above Tice with temperature fluctuations at the observation point are primarily concentrated in and around the Antarctic**

**Peninsula and the Weddell Sea in the Antarctic and encompass the east coast of Greenland and Northern Scandinavia in the Arctic."**

- *L. 319: "... suggests a correlation with orographic waves with ice PSCs above Tice". That ice PSC are correlated with orographic waves is a given from the beginning – you even explain the mechanism in the introduction. You could say that your results are a strong confirmation of these correlations.*

  **Based on the comments by one of the other referees we omit using the term "correlation" and rephrased the sentence as follows: "The ice PSCs above Tice with temperature fluctuations along the backward trajectories are primarily concentrated over the Antarctic Peninsula and the Weddell Sea in the Antarctic and encompass the east coast of Greenland and Northern Scandinavia in the Arctic, which are known hotspots of mountain wave activity."**


*Summary:*
*The manuscript uses a decade of MIPAS satellite observations of PSCs (polar stratospheric clouds) to investigate the water ice PSC occurrence with respect to frost point temperature, determines their spatial location, and examines back trajectories in order to understand the temperature history of the air parcels in which PSCs are observed. The authors consider the role which small-scale temperature fluctuations (i.e. gravity waves) play in PSC formation and quantify the fraction of PSCs associated with orographically-forced gravity waves. Because of the limb-viewing geometry of the MIPAS satellite, there are uncertainties in the position of the clouds along the line of sight observation path. The authors propose using the point where temperatures are closest to the frost point as the location of the ice PSCs.*

*The manuscript is well written and provides plenty of evidence to support the idea that orographic gravity waves can play an important part in driving ice PSC formation. I think that the manuscript should become acceptable for publication once the authors consider implementing the below comments.*

**We would like to thank the referee for the encouraging statement.**

*Major comment:*
*Overall comment: Most of the manuscript builds up to the point that orographic gravity waves can explain a percentage of the ice PSC observations (33% in the Arctic, 9% in the Antarctic, as noted in the abstract). Yet it seems to me that this is already a well-established result in the literature (much of which is cited within the text already) and so these parts of the manuscript are really just supporting that which is known already. I feel that the novel parts of the manuscript, i.e. the use of MPTRAC back trajectories together with the proposition to use air parcel history (in particular, minimizing the temperature difference to the frost point) as the location of the PSCs, should be brought forward as the central point of the manuscript, and thus emphasized much more clearly in the abstract and conclusions, and indeed through the whole text. Your title could also be reworded in this context.*

**Thank you for your comment. Based on your comment and the comments of the other two referees, we have revised and tried to improve the entire manuscript and hope that the novel parts of our study, including the method to identify the location of PSCs along the MIPAS line of sight and the use of Lagrangian backward trajectory analyses for MIPAS ice PSC detections, are now clearer. Please see the responses to specific comments below. In addition, we have changed the title to: "Impact of mountain-wave induced temperature fluctuations on the occurrence of polar stratospheric ice clouds: A statistical analysis based on MIPAS observations and ERA5 data".**

*Line 76 onwards: Discussion about method of identifying most likely ice PSC location as minimiz-*

*ing relative to the frost point temperature. Does this method reduce the hundreds of kilometers of horizontal uncertainty (which you noted on line 80)? By how much? I should like to see some attempt to quantify the reduction in error as I think that is central to your manuscript. For instance, with the method here do you observe any gravity-wave connected ice PSCs upwind of mountain ranges? This may help indicate a bound on your accuracy. You would likely have better ideas on quantifying this uncertainty.*

**Since the real location of a PSC along the MIPAS line of sight remains unknown as ground truth, it is difficult to quantify the uncertainty and improvements in locating the PSC detections with different methods. Assigning the detection to the tangent point, as in previous studies, is a heuristic choice. Similarly, assigning the detection to the location of the minimum of the temperature difference between Tice and T along the line of sight (△Tice_min) is also a heuristic but likely more plausible choice.**

**In Sect. 4.2 of the paper, we discuss the sampling uncertainty of MIPAS for the ice PSC detections. In particular, we evaluated how the choice of different methods of locating the PSCs (tangent point versus △Tice_min) affects the spatial and temporal distributions of the observed occurrence frequencies (compare Figs. 1 and 9, respectively). We found a larger sensitivity to the height difference than to the horizontal distance of the tangent point and the location of the △Tice_min minimum, which we attribute to the fact that temperature changes significantly with height but not so much horizontally. In the statistical comparison with $3° \times 5°$ horizontal grid boxes conducted here, the horizontal distribution of ice PSCs is found to be essentially identical for both methods.**

**Although we cannot quantify the uncertainty of the individual locations of the MIPAS PSC detections due to the lack of ground truth information, we performed an additional statistical analysis to determine the horizontal and vertical distances between the tangent point and the temperature difference minimum along the line of sight (see Fig. 1 in this reply). This analysis shows that half of the time the horizontal distances are below $\pm$ 200 km and the vertical distances are below 4 km. This is consistent with the discussion in Sect. 4.2 of the paper.**

[Figure]

Figure 1: Histograms of the horizontal (left) and vertical (right) distances between the tangent point and the minimum of the temperature difference between $T_{\mathrm{ice}}$ and $T$ along the line of sight. The statistics cover MIPAS observations during Southern Hemisphere polar winter in the years 2003 to 2011.

*Minor Comments:*
*Line 1: 'occurrence of ice clouds'. I think you mean PSC water ice clouds, rather than tropospheric cirrus clouds. Clarify.*

**Yes, we mean polar stratospheric ice clouds. We have rephrased the sentence.**

*Line 2: which decade?*

**The data considered is from 2002-2012. We have added the years in the sentence.**

*Line 13: 'correlation': have you performed a correlation analysis? I couldn't find one. Correlation doesn't imply causation anyway.*

**Yes, you are right. We have not performed a correlation analysis. Using this term here was a bit misleading. We have rephrased the sentence.**

*Line 24 (or somewhere close by): Given that you spend much of manuscript discussing 'small-scale temperature fluctuations', I think you need to introduce the concept of gravity wave generation by mountains, explaining that these can cause small horizontal scale temperature perturbations in the stratosphere which are large enough to enable some PSC formation in the cold phase of these waves, even when the background (synoptic) scale temperature is too warm for their formation. And thus missing these waves in model (climate) simulations will effect the model PSC representation. Plenty of references abound for this discussion, many of which you already cite.*

**Following also the comments of the other referees, we added a new paragraph to summarize the background and assumptions of this study more clearly: "Atmospheric gravity waves are oscillations in the Earth's atmosphere caused by buoyancy and gravity acting on air parcels displaced from their equilibrium positions. They can significantly affect local pressure, temperature, winds, and other meteorological variables. Gravity waves play a critical role in the transfer of energy and momentum between different layers of the atmosphere, i. e., they can significantly affect the temperature structure and general circulation of the atmosphere (Fritts and Alexander, 2003; Alexander et al., 2010). Gravity waves typically originate from sources such as mountain ranges, where air flow is disturbed and lifted; thunderstorms, where convective activity is generated; and frontal systems, where air masses are disturbed from geostrophic equilibrium. Understanding gravity waves is essential for accurate weather forecasting and broader atmospheric dynamics. Properly representing the effects of gravity waves in general circulation and chemical transport models is challenging because coarse-grid global models typically lack the spatial resolution needed to resolve the small-scale perturbations of gravity waves. In this study, we are particularly interested in atmospheric gravity waves in the polar winter lower stratosphere, because temperature perturbations due to gravity waves can trigger the development of PSCs in the cold phase of these waves, even in cases where background or synoptic-scale temperatures are higher than their formation thresholds (Carslaw et al., 1998; Rivière et al., 2000; Dörnbrack et al., 2020; Orr et al., 2020). Missing these waves in coarse-resolution global chemistry-climate simulations or reanalysis-driven chemistry-transport simulations will affect the PSC representation."**

*Line 38 (or somewhere close by): You haven't discussed the CALIOP wave-ice PSC class, as defined by Pitts et al. 2013 doi:10.5194/acp-13-2975-2013. This class addresses a similar problem*

*to what you are here, namely using satellites to attempt to quantify gravity wave influence on PSC formation, but in a different way to how you do it with MIPAS. Some mention of the CALIOP wave-ice would be beneficial in the introduction.*

**We agree and added the following sentence referring to the CALIPSO wave ice PSC class introduced by Pitts et al. (2013): "Due to the importance of wave generated ice PSCs, Pitts et al. (2013) introduced an additional class in their PSC characterisation scheme specifically for wave ice PSCs."**

*Line 51: 'we explore the temperature variations': There seems a bit of a reluctance in the paper to use the phrase 'gravity waves' or 'orographic gravity waves' when this is what you are doing. Maybe because you are building up to attributing the ice PSCs to gravity wave activity. Consider rephrasing. See also Major Comment above.*

**We agree that the relation of PSC formation to gravity wave activity needs to be more clear and explicit throughout the paper. In addition to changes in other places, we rephrased: "...we explore the temperature variations along the backward trajectories driven by ERA5 data as an indicator of gravity wave activity to comprehend their impact on ice PSC occurrence."**

*Line 85: You use ERA-Interim for this identification but use ERA5 elsewhere, so why use ERA-Interim anymore? There is no explanation given, so please provide a justification. But, line 116 says ERA5 is much better for Lagrangian transport, it is "significantly impacted" as you write. So I think that you should really use ERA5 for identifying the minimum difference to the frost point. It will capture the small-scale gravity waves more effectively than ERA-Interim, which is a major point of your study. Also given the finer resolution in ERA5, I would expect that it will improve your results and likely reduce the horizontal uncertainties. See also my Major Comment 'Line 76 onwards' above.*

**Thank you for pointing this out. After rechecking the data, we can confirm that only ERA5 was used in our study. The data set we use here is the one described in Spang et al. (2018). In the original dataset provided with the paper, the respective ERA-interim temperatures along the line of sight were included in addition to the MIPAS PSC data. However, in updates to this MIPAS PSC data, the data was provided with ERA5 temperatures. Our manuscript was based on one of the older versions, and when we updated the MIPAS PSC dataset, we accidentally missed updating the text in the manuscript. We apologize for this oversight and appreciate that this mistake was discovered through the referees' questions.**

*Line 88: 'up to -6.1km below' reads awkwardly. Reword and simplify.*

**We have revised to "at an altitude of 6.1 km below the cloud top".**

*Line 123: Add year to Koop reference*

**The year has been added.**

*Line 129: 'less than 10K above' should be reworded for simplicity*

**We have changed our wording to "within 10 K above Tice"**

*Line 139: 'frequency over 16%' – but your scale is cut at 16% so you can't comment on this!*

**We have changed our wording to "frequencies up to 16%"**

*Line 142: discussion of the Arctic vortex. I think you need some references and explanations about the off-pole position of the Arctic vortex here, noting the role of planetary waves in the NH in forming the vortex in this location.*

**We have added the following sentence including references: "Note that while the Antarctic polar vortex is centred over the pole, the Arctic polar vortex is displaced from the pole due to frequent disturbances from sudden stratospheric warmings (SSWs) caused by planetary wave activity (Waugh and Randel, 1999; Baldwin et al., 2021)."**

*Line 143: A sentence linking this climatology to previous PSC climatologies would be useful (e.g. see Tritscher et al. Rev Geophys 2021, their Fig 11 and Fig 21)*

**Directly comparing our Fig. 1 with Fig 11 and 21 from Tritscher et al. (2021) is not possible since we show in Fig 1 a multi-year (2002-2012) averaged occurrence frequency while Fig. 11 and Fig 21 from Tritscher et al. show the occurrence frequency for one specific year (2009) and separated by month for averaged over the time period 2006-2018, respectively. Nevertheless, since the general features are the same (spatial location of the PSCs and the descending height of the PSCs during the course of the winter), we changed the sentences describing Figure 1 as follows: "The spatial and vertical distribution of the multi-year averaged occurrence frequency of ice PSCs at $\triangle$Tice_min from MIPAS (2002-2012) is presented in Fig. 1. The results of the spatial and vertical distribution of ice PSCs derived here are in general agreement with the respective occurrence frequencies from MIPAS and CALIPSO shown in Tritscher et al. (2021)."**

*Line 148: sentence beginning 'Since ice PSCs...' this sentence on thresholds was discussed already. Consolidate or remove.*

**We have consolidated the duplicate parts here.**

*Line 150-151 : You should consider a cumulative frequency plot to illustrate the points you are making here (i.e. the 90% and 100% thresholds)*

**Thanks for the suggestion. The cumulative frequencies can be interpreted from the two vertical dashed lines at Tice-1.5 K and Tice-3 K in Fig. 2.**

*Line 151: 'T-Tice-3K' recheck this phrase*

**Checked and corrected.**

*Line 162: Worth stating clearly that these are low fractions (a few %). But, they are similar to CALIOP results (e.g. Alexander et al. 2013).*

**We have added one sentence. "Low fractions (about 1%) of ice PSCs are present in the Arctic which are similar to CALIOP results (Alexander et al., 2013)."**

*Line 167: Years 2005 and especially 2011 were major SSWs with large ozone losses (See Manney et al. Nature, 2011, doi:10.1038/nature10556). Think about the larger context, and suggest*

*you cite this Manney paper, or others which you may wish, to provide this larger overview of the interannual variability.*

**We agree. We cite now Manney et al. (2011) for the Arctic winter 2010/2011 and Feng et al. (2007) for the Arctic winter 2004/2005 and added the following sentence in the manuscript: "The Arctic winters of 2004/2005 and 2010/2011 were both distinguished by exceptionally low temperatures and substantial ozone depletion (e.g. Feng et al., 2007; Manney et al., 2011)."**

*Figure 4: You should add some uncertainty bars on these panels. You have 10 years of data. These can describe the interannual variability*

**The uncertainty bars have been added.**

*Figure 6: A really bad color choice of a red star! I suggest you use black*

**Thanks, we have updated the plot.**

*Section 3.5 title and wording 'Correlation'. Are you actually performing a correlation analysis? Seems not. I think you need to remove this word 'correlation' throughout your text.*

**We do not perform correlation analyses. We have adjusted the section title and omitted using the term "correlation".**

*Line 222: Is gravity wave activity really more frequent in the NH (mountainous regions) than in the SH? You need to provide a reference here! Or, is it that the Arctic is synoptically warmer and generally hovers just above Tice, so any orographic waves are more likely to increase ice PSC fraction when compared with the synoptically colder Antarctic. Explanation is needed, either way.*

**Orographic waves are not more frequent in the southern hemisphere, but these have a larger impact on PSC formation in the Arctic since temperatures are generally warmer there and do not reach by synoptical cooling alone so often temperatures that are sufficiently low for ice particle formation (e.g Tritscher et al., 2021). We have rephrased the sentence as follows to make this more clear: "This difference can be attributed to the higher importance of gravity wave activity on ice PSC occurrence in the Northern Hemisphere."**

*Line 237: Can you quantify, or at least state/cite the underestimation of temperature fluctuations?*

**We added: "Note that while meteorological reanalyses are often capable of reproducing gravity wave events at the right time and place, especially for mountain waves, the wave amplitudes in the reanalyses are typically damped compared to the real atmospheric conditions (Schroeder et al., 2009; Jewtoukoff et al., 2015; Hoffmann et al., 2017). The degree of underestimation depends on the resolution and numerical filters of the forecast model used to generate the reanalysis and on the spectral characteristics of the gravity waves."**

*Line 238: 'The analysis applied in this study provides important information on the frequency and significance of these discrepancies': I'm afraid that I can't find the text which supports this statement. Please refer back to the figure / section where this is true, or remove/reword.*

We have rephrased the sentence as follows: "The analysis applied in this study (see previous sections) provides important information on the frequency of these events and once again points out the significance of the discrepancies between observed ice PSCs and temperatures derived from meteorological reanalysis."

*Line 262: Uncertainty of approximately 0.2K. Is this about the figure for the polar regions? How bad is a 0.2K uncertainty? Your bins in Figure 10 are much larger so I suspect this uncertainty is not a big deal. Comments required.*

The 0.2 K is the global mean temperature uncertainty in the low and middle stratosphere according to Simmons et al. (2020). Unfortunately, the temperature uncertainty in the ERA5 data over the polar stratosphere is unknown.

*Line 267: You write that you use water vapor data from ERA5. But you can obtain water vapor data from MLS satellite (e.g. see Tritscher et al. 2021, Rev. Geophys.). Have you considered doing so?*

Yes, we have considered using water vapour from MLS, but decided against it. If we would use MLS observations together with MIPAS we would have to deal with the uncertainties of two satellite instruments and measurements made in two different orbits. Further, the MLS measurements start from 2004, thus two years later than MIPAS and thus do not cover the same time period. ERA5 reanalysis data on the other hand is globally available for the entire MIPAS measurement period. Additionally, we think that also for calculating Tice along the trajectories using ERA5 $H_2O$ is more appropriate despite the uncertainties.

*Line 291: How about also considering the actual generation mechanism of the waves? Dornbrack et al. (2001, doi: 10.1029/2000JD900194) provide a set of criteria for mountain wave activity. Essentially, at many times, near-surface winds are not conducive for the formation of orographic gravity waves. They only happen when surface winds are of the correct angle against the mountain range, and the background wind allows their propagation into the stratosphere. So while it may be the 'unresolved temperature fluctuations in ERA5' which is the issue, or it could be something else.*

Indeed, thanks for mentioning this. However, the generation mechanism of waves is beyond the scope of this study. We have added one sentence in the manuscript "Other reasons, such as the formation mechanism of gravity waves (Dörnbrack et al., 2001), could also affect the results, but taking these into account is beyond the scope of this study."

*Line 295: 'are proposed to' – what about 'are shown to'?*

They have both proposed and shown to provide a better estimate. Thus, we changed the sentence as follows: "The points with the smallest temperature difference ($\triangle$Tice_min) between the frost point temperature (Tice) and the environmental temperature along the line of sight have been proposed and shown to provide a better estimate of the location of ice PSC observation from MIPAS."

*Line 300: 'which is about 2-4 km higher above the tangent point'. This phrase needs explaining*

**Revised as "which is about $2-4$ km higher than that detected at the tangent point"**

*Line 316: where has the 'significant correlation' been demonstrated?*

**See our answer to your comment above. We have omitted using the word correlation.**

*Line 324: As suggested above, what about the point that the Arctic is synoptically warmer and thus orographic waves' influence on PSC formation can have a larger fractional impact?*

**Agree, we rephrased this text part as follows: "This difference can be attributed to the larger role of gravity wave activity in ice PSC occurrence in the Northern Hemisphere. This is due to the generally warmer temperatures in this hemisphere, which often require mountain wave-induced temperature fluctuations to initiate ice PSC formation. However, the larger size of the selected region may also contribute to the hemispheric differences."**


**We have carefully checked the manuscript and improved the grammar and removed all mistakes in the manuscript.**

Minor comments
*1 Line 1: The opening line on the importance of mountain waves as a PSC formation mechanism*

*seems completed disconnected with most of the abstract – especially the earlier parts. This needs to be much more coherent.*

**Yes, we agree. We revised the beginning of the abstract as follows: "Temperature fluctuations induced by mountain waves can play a crucial role in the formation of polar stratospheric clouds (PSCs). In particular, the cold phase of the waves can lower local temperatures sufficiently to trigger PSC formation even when large-scale background temperatures are too high. To provide new quantitative constraints on the relevance of this effect, this study analyzes a decade $(2002 - 2012)$ of ice PSC detections obtained from Michelson Interferometer for Passive Atmospheric Sounding (MIPAS/Envisat) measurements and ERA5 reanalysis data in the polar winter lower stratosphere."**

*2 Line 3: The definition of $\Delta$ Tice_min is poorly described and confusing.*

**We have removed this sentence in the abstract and the detailed definition of $\Delta$Tice_min can be found in Sect. 2.1.**

**"Therefore, instead of using the tangent point of the sample, we employ the point with the minimum temperature difference ($\Delta$Tice_min ) between the frost point temperature (Tice) and the environmental temperature along the line of sight ($\Delta$Tice_min = min(TLoS − Tice)), up to a maximum altitude of 30 km, to identify the most probable position of the ice PSC observation."**

*3 Line 27: I would quantify what you mean by small-scale waves, i.e., explicitly give the scale in km.*

**We do not refer here to small-scale waves, but to small-scale temperature fluctuations. These can decrease the temperature by several degrees so that the temperature threshold for solid PSC formation, e.g. ice, are reached. These temperature fluctuations and their amplitudes cannot be resolved by the ERA5 grid.**

*4 Line 30: This sentence is not quite clear. Are you saying that 63% of all ice PSCs during the period of interest were associated with mountain waves? Please clarify.*

**Yes, Noel and Pitts (2012) analysed CALIPSO data covering the time period from 2006-2010 and found that 63 % of ice PSC were observed during GW events.**

*5 Line 41: Grammar. Suggest 'Carslaw et al. (1999) unveiled'.*

**We changed this to "have unveiled"**

*6 Line 44: Maybe better to refer to 'mountain waves' throughout the text, rather than swop between gravity waves and mountain waves.*

**Although most case studies focus on gravity waves from orographic sources, Hitchman et al. (2003) and Shibata et al. (2003) showed that non-orographic gravity waves can also trigger PSC formation. Nevertheless, we have reviewed the text and carefully checked that the terms "gravity waves" and "mountain waves" are used appropriately.**

*7 Line 43: I'm not quite sure what the sentence beginning 'Temperature perturbations . . . ' is trying to say here.*

**We removed this sentence.**

*8 Line 44: The waves are 'fine-scale' temperature fluctuations, not 'subgrid-scale'. And its not 'may not be fully resolved', its 'are underestimated / not fully resolved'.*

**Yes, you are right. The sentence has been revised.**

*9 Line 61-62: Slightly awkward sentence. Can it be revised.*

**We revised the sentence as follows: "The Michelson Interferometer for Passive Atmospheric Sounding (MIPAS) on board the Envisat satellite measured limb infrared spectra in the $4 - 15$ $\mu$m wavelength range with high resolution from the mid-troposphere to the mesosphere."**

*10 Line 68: The CI needs to be defined first, before a statement can be made saying its sensitive to PSCs. This paragraph maybe needs some reorganisation.*

**We added the following paragraph introducing the cloud index method: "Spang et al. (2001) introduced a simple and reliable method for detecting clouds in infrared limb sounder measurements by comparing the mean radiances of two different spectral wavelength regions. Each region responds differently to clouds in the field of view. The first region, $788 - 796 \, \mathrm{cm}^{-1}$, is primarily influenced by $CO_2$ emissions and shows little change in the presence of optically thin clouds. In contrast, the second region, $832 - 834 \, \mathrm{cm}^{-1}$, is located in an atmospheric window region and mainly influenced by aerosol and cloud emissions. The ratio of radiances, called the cloud index (CI), is high for cloud-free conditions (CI $> 4$), close to one for optically thick conditions, and falls in between for the transition from optically thin to thick clouds."**

*11 Line 74: ICE_NAT, STS_NAT etc need to be properly explained/defined, ie state that class #4 is a mixture of ice and NAT PSCs.*

**We have added the following sentence: "Class 1-3 are pure type classes, while class 4-6 are mixed type classes.".**

*12 Line 86: Please see comment above about the CI not being properly defined. This needs to be done so that the reader understands what CI > 1.2 means.*

**We tried to clarify this with the reply to comment 10 on line 68.**

*13 Line 47 and 85: Not sure whether its ERA-Interim or ERA5 being used. Please clarify. Title says ERA5. Also, need some explanation of how reliable reanalysis is for this task.*

**Only ERA5 data has been used in this study as clarified in our answers to your comments 3 and 4. The introduction of ERA5 data can be found in Section 2.3 and its temperature uncertainty has been discussed in Section 4.3.**

*14 Line 95: MPTRAC already defined.*

**Correct. We have omitted defining MPTRAC here once again.**

*15 Line 98. Grammar. I think this should 'e.g.' and not 'i.e.'.*

**We changed "i.e." to "e.g.".**

*16 Line 99: FLEXPART needs to be defined.*

**Done.**

*17 Line 107 to 111: This paragraph should be included when ERA-Interim is first used in the methods. The heading for section 2.2 should also be revised.*

**This paragraph has been removed and the title of subsection 2.2. has been adjusted.**

*18 Line 120: Repetition. It was only just mentioned earlier that ERA5 was used to calculate Tice.*

**Sentence has been rephrased so that this repetition is omitted.**

*19 Line 134: This sub-heading does not seem appropriate. Is only 3.1 focused on MIPAS?*

**Also 3.2 focused on MIPAS data. We have adjusted the subsection title of 3.1 to "Ice PSC observations".**

*20 Line 135: Repetition. This has already been explained in the methods. And if wasn't completely explained there, then absolutely should be.*

**Indeed we have already explained this, but think that this repetition here does not matter.**

*21 Line 138: Please refer to latitude and longitude correctly. 'south of 65deg' means nothing, unless its written 65S. Also 'longitude range of +-90deg' is also wrong. Please ensure that lat and lon are correctly described everywhere in the text.*

**Done.**

*22 Line 139: Grammar. Highest occurrence frequency over 16%. Also 'Over the seasons' – not sure what that means.*

**Sentence has been corrected and "Over the seasons" has been changed to "Over the course of the year"**

*23 Line 145: Sentence beginning 'In both polar regions' does not make sense.*

**This sentence has been removed due to the comment by referee 1.**

*24 Figure 1: Colour bars are not labelled. The caption does not mention what the different panels are. Why are different font sizes used for the labels in panels b and d? Presumably panels b and d show mean values over the entire polar region? - This is not clear from the caption. In the figure, I don't think that the labels for each panel are necessary or even helpful.*

**The left and right figures have the same colour bar and the labelling is solely done on the panels on the right side. The panels a and b show the occurrence frequency for the Antarctic and panels c and d for the Arctic. What the different panels show has been added to the figure caption.**

*25 Figure 1: This is unclear: difference between Tice and T along the line of sight ($\Delta Tice\_min$). Why have such a convoluted way of explaining what Tice-T is? Also, it should be written as*

$\triangle Tice\_min = T - Tice$ (ie as a formula).

**We consider here the minimum difference between the temperature at the line of sight TLoS and Tice, thus $\triangle$Tice_min = min(TLoS - Tice). We have added the formula in the text.**

*26 Lines 144-145: Convoluted way of explaining what is in Fig. 2. Either use T-Tice everywhere, or defined $\triangle Tice = T - Tice$ and use this everywhere.*

**Please see our response to major comment 5. $\triangle$Tice_min and T-Tice are not the same. $\triangle$Tice_min is used to derive a specific location where the temperature difference is at its minimum, indicating the presence of ice PSCs. T - Tice is a general expression for temperature difference.**

*27 Figure 2: Same comments as above. Figure does not have units of temperature. The label SH $\triangle$ Tice_min is wrong as the axis is already labelled as T-Tice. The panel labelling of 'ICE' is unnecessary. Also, not sure what 'Fraction to observations' means.*

**We have improved the figure. Please find the explanation of the difference between $\triangle$Tice_min and T - Tice in response to major comment 5.**

*28 Line 147: Grammar. Comma after 56% not required.*

**Correct. The comma has been removed.**

*29 Line 148: 'Derived from ERA5 reanalysis' not required.*

**Fixed.**

*30 Line 149-150: Repetition. T-3K and T-1.5K are already explained.*

**We have consolidated the duplicate parts.**

*31 Line 150: Is T-Tice-3K correct? Should it not be Tice-3K? Why is a ')' included here.*

**It should read Tice-3K. We have corrected this and removed the obsolete parenthesis.**

*32 Line 154: Please revise sub-headings. I'm not sure how the material in this sub-section differs from the previous section.*

**Both sections deal with the same data set, namely MIPAS, but these differ, because in 3.1 we describe the MIPAS ice PSC observations in general and in section 3.2 we focus on the ics PSCs that were observed at temperatures above Tice and their characteristics. We have revised the sub-header of 3.1 to "Observation of ice PSCs"**

*33 Figure 3 caption: Grammar – please correct. Also, normally captions say 'Analogous' rather than 'Similar' in this context.*

**We changed "Similar to" to "Same as".**

*34 Line 160: Not sure use of 'trend' is appropriate here. Could simply say 'decrease in altitude throughout winter'.*

**We agree and rephrased the sentence to omit the term "trend".**

*35 Line 161: This is a strange sentence, and does not make sense. Also, why is the physically basis for this rather randomly explained here, but not for other results?*

**The sentence has been revised. In the frame of the revision we have added explanations on the physical basis at several places.**

*36 Line 165: Rather than writing just 'comparable', you also need to give the values.*

**Values have been added. "the fraction of ice PSCs above Tice is 51 % in January and February, and 70 % in December due to less ice PSC observations."**

*37 Line 167-168: This sentence makes no sense. 'lower occurrence of ice PSCs . . . due to few ice PSCs'.*

**The Arctic winter 2004/2005 and 2010/2011 were both exceptionally cold winters. Thus, the observed ice PSCs during these winters occurred rather at temperatures below Tice than above Tice (i. e. during these winters ice PSCs were rather formed by synoptical cooling than by mountain wave induced cooling by temperature fluctuations). We have rephrased the sentence.**

*38 Line 169: Again, the explanation for this points is welcomed for its insight, but should be done consistently, or saved for the discussion. Broad statements such as the stability of the Antarctic vortex v Arctic vortex should be introduced in the Introduction – also surely results such as this have been readily explained / shown elsewhere.*

**With the sentences we added on L142 the differences between the Arctic and Antarctic vortex should be now more clear.**

*39 Line 172: Poor English.*

**The sentences have been corrected.**

*40 Figure 4: Please improve formatting of plots, such as the values on the axis. These are different for panels b and d, despite both panels being identical.*

**Thanks, we have improved the figure.**

*41 Line 175: What does 'point of observation at $\Delta$ Tice_min' mean?*

**$\Delta$Tice_min is used to pinpoint the exact location of ice PSCs. We revised this sentence to "we employed the MPTRAC model to calculate 24-hour backward trajectories from the ice PSCs observed at $\Delta$Tice_min ."**

*42 Line 176: '(t) t'*

**This has been corrected.**

*43 Line 179: These are not a 'trend'. Please use a different word.*

**We changed "trend" to "behaviour".**

*44 Figure 5: Not sure what label of vertical axis means. Caption is also far to brief and not enough information for the reader to understand the plot.*

**We have improved the figure and caption.**

*45 Lines 184-188: This text should be in the methods section. Why is the methods being explained in the results section?*

**We have removed the text from the result section.**

*46 Line 191: What is 't=-0'?*

**The minus is obsolete and has been deleted.**

*47 Line 193: $h^{-2}$ is not a rate. Rate is per hour.*

**This is not the rate, but the variance. Please note that we corrected the units of the cooling rate variances to $K^2\,h^{-2}$ throughout the manuscript.**

*48 Figure 6 caption: This seems to be written by a different person as the previous captions were brief. But surely no need to define T for temperature at this stage of the paper.*

**We agree that that temperature at this stage of the paper does not to be introduced and thus we removed "(T)".**

*49 Line 202: Confused here, as introduction stated that ERA5 poorly resolves temperature fluctuations (fine-scale) but here says that it does.*

**We agree that this sentence was misleading. We rephrased: "This suggests that the time and location of temperature variations in ERA5 can still be related to the presence of ice PSCs above Tice, especially in the Arctic, even if we consider that gravity wave amplitudes may be underestimated."**

*50 Line 228: What does warm large spatial scales mean?*

**We have rephrased the sentence as follows: "The occurrence of ice PSCs in warm environments have already been reported in previous studies".**

*51 This claim that ERA5 misrepresents fine-scale waves has never been justified in the paper. What study are you referring to?*

**In the introduction, we pointed out: "However, the small-scale temperature fluctuations related to mountain waves are often underestimated or not fully resolved in global reanalyses or coarse-resolution chemistry-climate models (Orr et al., 2015; Hoffmann et al., 2017; Orr et al., 2020; Weimer et al., 2021)." We have added here the reference of Orr et al. (2015), as they also discuss the difficulties of properly representing the effects of mountain waves on PSC formation in numerical simulations.**

**The study by Hoffmann et al. (2017, see Sect. 4) evaluated gravity wave variances from ECMWF IFS operational analyses from T511 (39 km effective resolution) to T1279 (16 km) in comparison with Atmospheric InfraRed Sounder (AIRS) satellite observations, and showed that the IFS simulations largely (up to a factor of ∼6) underestimated the**

**measured variances.**

**A new study by Lear et al. (2024) provides to our knowledge the first quantitative analysis of the representation of explicitly resolved gravity waves in ERA5 based on comparison with AIRS satellite measurements. The study supports the conclusion that gravity wave amplitudes are underestimated in ERA5.**